# Highly structured homolog pairing reflects functional organization of the *Drosophila* genome

Jumana AlHaj Abed [1,9], Jelena Erceg[1,9], Anton Goloborodko [2,9], Son C. Nguyen [1,6], Ruth B. McCole[1], Wren Saylor [1], Geoffrey Fudenberg [2,7], Bryan R. Lajoie[3,8], Job Dekker [3], Leonid A. Mirny [2,4] & C.-ting Wu[1,5]

*Trans*-homolog interactions have been studied extensively in *Drosophila*, where homologs are paired in somatic cells and transvection is prevalent. Nevertheless, the detailed structure of pairing and its functional impact have not been thoroughly investigated. Accordingly, we generated a diploid cell line from divergent parents and applied haplotype-resolved Hi-C, showing that homologs pair with varying precision genome-wide, in addition to establishing *trans*-homolog domains and compartments. We also elucidate the structure of pairing with unprecedented detail, observing significant variation across the genome and revealing at least two forms of pairing: tight pairing, spanning contiguous small domains, and loose pairing, consisting of single larger domains. Strikingly, active genomic regions (A-type compartments, active chromatin, expressed genes) correlated with tight pairing, suggesting that pairing has a functional implication genome-wide. Finally, using RNAi and haplotype-resolved Hi-C, we show that disruption of pairing-promoting factors results in global changes in pairing, including the disruption of some interaction peaks.

[1] Department of Genetics, Harvard Medical School, Boston, MA 02115, USA. [2] Institute for Medical Engineering and Science, Massachusetts Institute of Technology (MIT), Cambridge, MA 02139, USA. [3] Howard Hughes Medical Institute and Program in Systems Biology, Department of Biochemistry and Molecular Pharmacology, University of Massachusetts Medical School, Worcester, MA 01605-0103, USA. [4] Department of Physics, Massachusetts Institute of Technology (MIT), Cambridge, MA 02139, USA. [5] Wyss Institute for Biologically Inspired Engineering, Harvard University, Boston, MA 02115, USA. [6] Present address: Department of Genetics, Penn Epigenetics Institute, Perelman School of Medicine, University of Pennsylvania, Philadelphia, PA 19104-6145, USA. [7] Present address: Gladstone Institutes of Data Science and Biotechnology, San Francisco, CA 94158, USA. [8] Present address: Illumina, San Diego, CA, USA. [9] These authors contributed equally: Jumana AlHaj Abed, Jelena Erceg, Anton Goloborodko. Correspondence and requests for materials should be addressed to L.A.M. (email: leonid@mit.edu) or to W.T.C.-t. (email: twu@genetics.med.harvard.edu)

Major hallmarks of chromatin organization include chromosome territories[1], compartments of active (A-type) and inactive (B-type) chromatin as delineated by the conformation capture technology of Hi-C, as well as chromosomal domains variably known as contact domains or topologically associating domains (TADs)[2–7]. These layers of organization encompass countless *cis* and *trans* interactions that determine the 3D organization of the genome. Among *trans* interactions, an important class includes those occurring between homologous (*trans*-homolog) as versus heterologous (*trans*-heterolog) chromosomes. Although long considered relevant only to meiosis, *trans*-homolog interactions are now widely recognized for their capacity to affect gene function in *Drosophila*, where homologs are paired in somatic cells throughout nearly all of development (reviewed by refs. [8–12]). What remains unclear is the global impact of such interactions and our ability to comprehensively understand the structure of paired chromosomes. To tackle this issue, we examine genome-wide maps of *trans*-homolog interactions in a newly established *Drosophila* hybrid cell line (PnM, XY diploid). In particular, by taking advantage of the parent-specific single nucleotide variants (SNVs) in this cell line, we provide a global assessment of different properties of homolog pairing, including how tightly apposed homologous chromosomes are and whether pairing is uniform across the genome. Furthermore, due to the sensitivity SNVs afforded our study, we assess how homolog proximity correlates with precision of alignment and with genome function.

*Drosophila* is the first organism in which *trans*-homolog interactions were implicated in gene regulation, as it is here that somatic pairing was discovered[13] and subsequently found to impact intragenic complementation (ref. [14] and reviewed by refs. [8–12]). Somatic pairing has now been associated with numerous biological phenomena across many species, including mammals, where pairing has been implicated in DNA repair, X-inactivation, imprinting, V(D)J recombination, and the establishment of cell fate (reviewed by refs. [9,11]). Pairing-dependent gene regulation, a well-recognized form of transvection, is among the best understood of biological phenomena associated with pairing (reviewed by refs. [8–12]). In *Drosophila*, transvection has been observed at many loci, suggesting that pairing may even function as a regulatory mechanism genome-wide[15–19]. Recently, this view has been supported by computational simulations of homolog pairing in *Drosophila*[20]. Thus, the question as to whether pairing can serve as a global mechanism for regulating and coordinating function, possibly facilitating transvection genome-wide, is drawing increasing attention.

For many decades, foundational studies documenting the impact of *trans*-homolog interactions on genome function have relied heavily on genetic approaches to infer pairing (reviewed by refs. [8–12]). Recent studies have also used live imaging[21] as well as fluorescent in situ hybridization (FISH) achieving super-resolution to visualize pairing of genomic regions as small as a few kilobases to as large as megabases, wherein a single signal in a nucleus was interpreted as the paired state and two as the unpaired state[22–25]. Chromosome conformation capture technologies, such as Hi-C, have also been implemented in investigations of pairing in yeast[26]. A recent study used read pairs representing interactions between identical Hi-C restriction fragments in a *Drosophila* cell line (Kc$_{167}$ cells, XXXX tetraploid) to tease out allelic interactions, such as between two homologs and between two sister chromatids[27]. This study reported an enhancement of allelic pairing in active genomic regions as well as an involvement of architectural proteins. In addition, consistent with the Cap-H2 component of condensin II being an anti-pairing factor[28] and Slmb being a negative regulator of Cap-H2[28–30], this study reported increased

and decreased allelic interactions, respectively, in the absence of these factors.

Here, we describe our work in examining the detailed architecture of pairing, using haplotype-resolved Hi-C to specifically target the pairing that occurs between homologous chromosomes. Haplotype-resolved Hi-C has been used to investigate *cis* interactions within mammalian genomes[31,32] (see Erceg AlHaj Abed, Goloborodko et al.[33] for additional references), and diploid homolog pairing in yeast[26] and, in our companion paper (Erceg, AlHaj Abed, Goloborodko et al.[33]), we developed a general methodology, called Ohm (Oversight of homolog misassignment), for applying this approach that ensures minimal misassignment of reads and high stringency in the detection of pairing. Applied to mammalian and *Drosophila* embryos, this approach demonstrated pairing in the latter to be genome-wide and also provided a framework in which to consider pairing in terms of precision, proximity, and continuity. We further revealed a potential connection between pairing and the maternal-to-zygotic transition in early embryogenesis.

In the current study, we shift our focus to the fine structure of somatically paired homologs and, to that end, take advantage of the greater homogeneity and higher pairing levels of *Drosophila* cell culture. In particular, we generate a diploid cell line from a hybrid cross and then apply haplotype-resolved Hi-C, allowing us to achieve a high-resolution map of homolog pairing. This approach reveals *trans*-homolog domains, interaction peaks, and compartments as well as variation in the structure and precision of pairing, documenting an extensive interspersion of tightly paired regions with loosely paired regions across the genome. Excitingly, we also find a strong association between pairing and active chromatin, compartments, and gene expression. Our findings demonstrate a comprehensive and detailed view of the structure of homolog pairing and resolve the long-standing question of whether pairing can bear a genome-wide relationship to gene expression.

## Results

**PnM cells are diploid and hybrid with high levels of pairing**. We began our study by crossing two strains of the *Drosophila* Genetic Reference Panel lines (057 and 439) that differ by ~5 SNVs per kilobase (kb) (Supplementary Table 1, ref. [33]) to generate 2–14 h old embryos that were homogenized to start a cell culture, which spontaneously immortalized and then was serially diluted to generate clonal cell lines (Fig. 1a; "Methods"). The clonal line used in this study, Pat and Mat (PnM), homogeneously expresses myocyte enhancer factor 2, suggesting it to be of mesodermal origin (Supplementary Fig. 1a, b). Karyotyping, in combination with homolog-specific FISH proved PnM cells to be male, diploid, and hybrid, with only chromosome 4 showing irregularities (Fig. 1b, c; "Methods"). This was promising, given that many cell lines are often aneuploid or polyploid. Finally, FISH analyses targeting two heterochromatic and three euchromatic loci confirmed high levels of pairing (Fig. 1d).

**Homolog pairing in PnM cells is highly structured**. We next performed haplotype-resolved Hi-C on PnM cells, an approach which separates, in silico, the read pairs into five categories: *cis*-maternal, *cis*-paternal, *trans*-homolog, *trans*-heterolog, and unresolvable by haplotypes. In this form of Hi-C, each of the two fragments of genomic DNA that are ligated together by virtue of their proximity in situ are assigned a parental origin based on the SNVs they carry, thus permitting researchers to distinguish Hi-C read pairs that represent *cis*-maternal, *cis*-paternal, *trans*-homolog (*thom*), and *trans*-heterolog (*thet*) interactions. By selecting only those read pairs with at least one SNV per side, we obtained

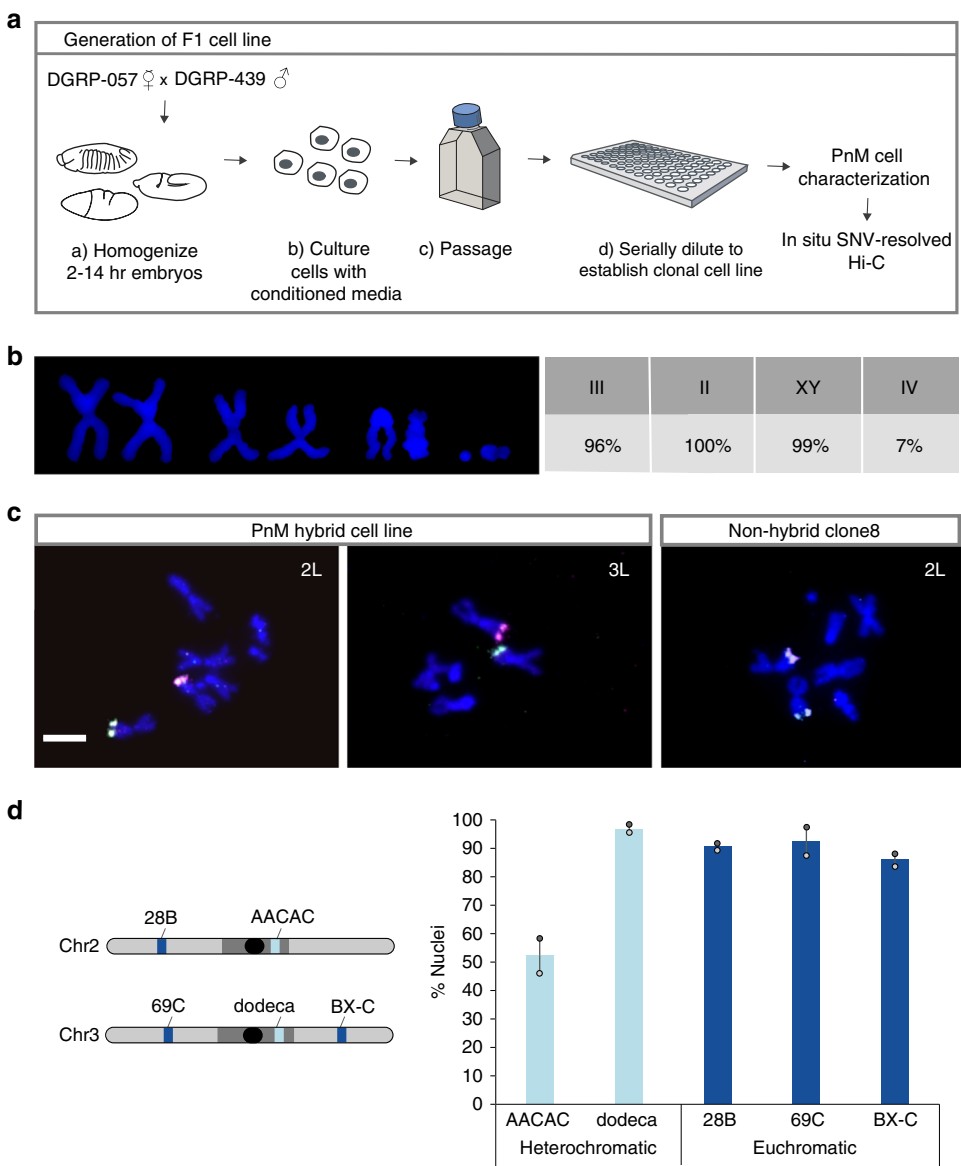

**Fig. 1** PnM cell line characterization. **a** Generation of the cell line. **b** Karyotyping demonstrates PnM cells to be male and diploid (N = 50). **c** Homolog-specific probes (HOPs) distinguishing 057-derived (magenta) from 439-drived (green) homologs for chromosomes (Chr) 2 and 3 on metaphase spreads for PnM and control (non-hybrid) clone8 cells confirmed PnM to be hybrid. Scale bar = 5 μm. **d** Left: Locations of heterochromatic (light blue) and euchromatic (dark blue) chromosomal regions targeted by FISH. Oval, centromere. Right: Levels of pairing in PnM cells quantified as percent of nuclei in which FISH signals, representing allelic regions, co-localized (center-to-center distance between signals ≤ 0.8 μm; error bars, s.d. for two biological replicates; N > 100 nuclei/replicate). Source data are provided as a Source Data file

75.4 million mappable read pairs, producing a 4 kb resolution haplotype-resolved map of the mappable portion of the genome (e.g., excluding repetitive regions), wherein less than 0.4% of *thom* read pairs are expected to have resulted from read misassignment ("Methods"; Supplementary Fig. 2a, b Supplementary Table 2). This gave us great confidence in our ability to select haplotype-specific reads, and then map them to the hybrid PnM genome.

As shown in Fig. 2a, homologs are aligned genome-wide, comparable to the global *thom* signature detected in early *Drosophila* embryos[33]. In addition, *trans*-heterolog interactions are detected globally and include sub-telomeric clustering (e.g., 2R to 3R)[34–36]. Strikingly, however, *thom* read pairs were ~7.8 times more abundant in PnM cells than in *Drosophila* embryos (Supplementary Fig. 2b). In addition, when considering *thom* contacts as a function of the separation of loci along the genome (genomic separation), we found them to be more abundant at all

genomic separations (Supplementary Fig. 2c, d). These observations are not surprising as they agree with the higher levels of pairing observed by FISH in PnM cells (Fig. 1d) as compared to developing embryos where pairing is just initiating[33,37], possibly due to a greater percentage of cells with paired homologs, an increased fraction of the genome exhibiting pairing, and/or a smaller proportion of dividing cells in the PnM cell line (Supplementary Fig. 3). Importantly, the greater abundance of *thom* contacts argued that an analysis of pairing in PnM cells would yield new insights into the structure of paired homologs.

**Homolog pairing can achieve high precision genome-wide.** We began by comparing the probability with which a locus will interact with another locus in *cis* as versus in *thom* at varying genomic separations (within a few kilobases and up to tens of megabases). We reasoned that, if pairing were maximally precise,

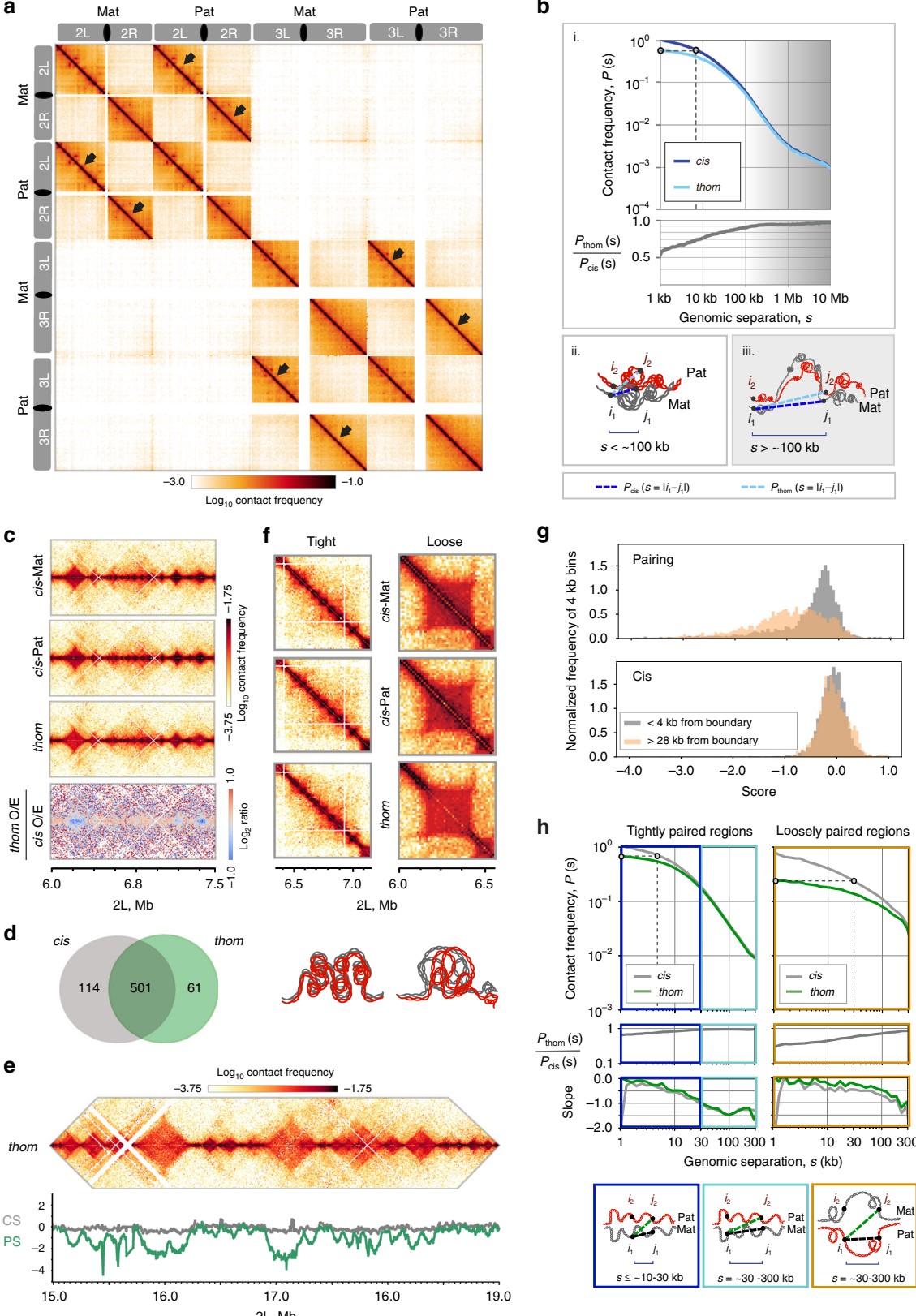

tight, and continuous ('railroad track'), our use of SNVs would reveal that any two loci would interact in *thom* nearly as often as they interact in *cis* regardless of genomic separation. In contrast, in the case of imprecise, loose, discontinuous pairing, the relative frequencies of *cis* and *thom* contacts could differ quite substantially. Note that this railroad track pairing does not preclude long-range *thom* interactions as paired chromosomes can fold back onto themselves, behaving as a single fiber.

**Fig. 2** Homolog pairing is highly structured, encompassing tightly and loosely paired regions. **a** Contact map for the left (L) and right (R) arms of chromosomes 2 and 3. **b**, i. Top, *cis* and *thom* contact frequencies, P(s), plotted against genomic separation, s, normalized to *cis* frequency at s = 1 kb. Dotted line: *thom* contacts at s = 1 were as frequent as *cis* at s = 8 kb. Bottom, $P_{thom}(s)/P_{cis}(s)$. *Cis* and *thom* contact frequencies differed noticeably at s < ~100 kb and were concordant at s > ~100 kb (shaded). **b**, ii–iii. Dashed lines, $P_{thom}(s)$ as a function of $s = |i_1-j_2|$ for loci $i_1$ and $j_2$ located on different homologs, compared to $P_{cis}(s)$ for loci $i_1$ and $j_1$ separated by $s = |i_1-j_1|$ on one homolog. Left, P(s) at s < ~100 kb. Right, P(s) at s > ~100 kb, shaded. **c** 1.5 Mb region on 2 L: *Cis* contact maps were concordant with each other and with the *thom* contact map. Bottom, *thom/cis* map showed concordance of *thom* with *cis* maps, apart from a depletion of contacts in some domains (blue). **d** Overlap of domain boundaries as defined by *cis* and *thom* contacts. **e** 4 Mb region of 2 L: Top, *thom* domains, and insulating boundaries. Bottom, pairing score (PS, green) and *cis* score (CS, gray). **f** Examples of tight and loose pairing with schematics of possible structures. **g** Distributions of PS and CS near (<4 kb, grey), or far from (>28 kb, orange) boundaries revealed higher pairing near boundaries. **h** Top, *cis* and *thom* contact frequencies, P(s), plotted against genomic separation, s, within tightly and loosely paired regions (normalized as in Fig. 2b). Middle, $P_{thom}(s)/P_{cis}(s)$; bottom, slopes. Tightly paired regions showed two modes of decay, shallow (dark blue box) and steep (light blue box), while loosely paired regions showed one shallow mode (orange box). Dashed lines: *thom* contacts at s = 1 kb were as frequent as *cis* contacts at s = ~5 kb and ~30 kb in tightly and loosely paired region, respectively. Schematics illustrate differences in the organization of tightly and loosely paired regions

To explore this line of reasoning, we calculated the genome-wide average contact frequencies for *cis* as well as *thom* contacts between pairs of loci, i and j (with i and j being on different homologs for *thom*) as a function of genomic separation (s = |i−j|) measured in base pairs (Fig. 2bi top): $P_{thom}(s)$ and $P_{cis}(s)$ ("Methods"). In fact, when the $P_{thom}(s)$, is plotted as a function of genomic separation, it is clear that *thom* contacts can occur at all genomic separations. Remarkably, $P_{thom}(s)$ peaks at the smallest genomic separation (s = 1 kb) in near perfect registration, with *thom* contacts being as frequent as contacts in *cis* at s = 8 kb. Moreover, the ratio $P_{thom}(s)/P_{cis}(s)$ is as high as 0.5–0.7 for genomic separations of 1–10 kb and only gets higher, approaching 1.0, with increasing genomic separation (Fig. 2bi, bottom). This high ratio indicates that a locus on one chromosome interacts with another locus nearly as often in *thom* as it does in *cis*, especially for s > ~100 kb (Fig. 2bi, iii, shaded). The simplest interpretation of this observation is that, overall, homologs are aligned in good register genome-wide, almost as a railroad track. We note, however, three caveats. First, our analysis only captured *thom* interactions that were accessible by Hi-C technology; for example, the repetitive regions of the genome were not mapped by Hi-C in either *thom* or *cis*. Second, our studies assume that *cis* and *thom* interactions are equally tractable. Third, as our Hi-C studies are a population assay, we cannot rule out cellular heterogeneity in the degree of pairing.

**Paired homologs form *thom* domains and compartments**. Besides the general distance-dependent decay of *cis* and *thom* contact frequency, our Hi-C maps also revealed a rich structure of *thom* interactions, including well-defined *thom* domains, at genomic separations as small as tens of kilobases, loops or interaction peaks, as well as plaid patterns of contacts far off the diagonal, at genomic separations as large as tens of megabases and corresponding to compartments (Supplementary Fig. 4a, Supplementary Fig. 5a). Consistent with railroad track pairing, we found strong concordance between the *thom*, *cis*-maternal, and *cis*-paternal Hi-C maps in terms of the positions and sizes of domains and loops (Fig. 2c, with Hi-C diagonal positioned horizontally; Supplementary Fig. 4b, d, Supplementary Fig. 5a), although with some exceptions (Supplementary Fig. 5b). In addition, 81.5% and 89.1% of the domain boundaries in the *cis* and *thom* maps appeared in the *thom* and *cis* maps, respectively (Fig. 2d). Overall, the strong concordance between *thom*, *cis*-maternal, and *cis*-paternal Hi-C maps indicated a high level of registration between paired homologs.

**Homolog pairing includes tightly and loosely paired regions**. Looking more closely at our *thom*, *cis*-maternal, and *cis*-paternal Hi-C maps, we discovered that some *thom* domains lacked a prominent signal along the diagonal (Fig. 2c, bottom panel showing subtraction Hi-C map; Supplementary Fig. 4d), suggesting an overall looser pairing. This absence of the diagonal contributed to the lower values of $P_{thom}(s)/P_{cis}(s)$ at genomic separations of s < ~100 kb (Fig. 2bi, ii unshaded) and clearly demonstrated that pairing was not uniform across the genome. We quantified this variation of pairing via a pairing score (PS), defined as the $\log_2$ average *thom* contact frequency near the diagonal, where both reads of a read pair lie within a ±12 kb window around a given 4 kb bin, and compared that score to an analogous score for *cis* contacts (CS) (Fig. 2e; "Methods"). Thus, as Hi-C data reflect the frequency with which interacting genomic regions colocalize, the PS served as a proxy in our analyses for the relative tightness or looseness of pairing across a chromosomal region. Figure 2e illustrates how dramatically the PS can vary along the chromosome, dipping most noticeably when the *thom* diagonal is lacking from the central region of a domain. In line with this and compared to scores for *cis* contacts, PS values for loci within 4 kb (one bin) of domain boundaries are overall much higher than those for loci greater than 28 kb from the nearest boundary, consistent with the lack of diagonals coinciding with the central regions of domains (Fig. 2g) and pointing to tighter pairing at domain boundaries. Lack of a diagonal may reflect any number of structures, including imprecise and/or loose pairing or even the side-by-side alignment of homologous, yet distinguishable, domains. In contrast, domains retaining the structure and diagonal observed in corresponding *cis* domains may represent railroad pairing throughout the domains (Fig. 2f, schematics below).

To better understand genome-wide variation in pairing, we examined the PS distribution (Supplementary Fig. 6a) and noted that it could be approximated by two normal distributions that were reproducible for each replicate (Supplementary Fig. 6b, d). These distributions suggested two classes of loci, one consisting of more tightly paired (higher PS) loci and the other consisting of more loosely paired (lower PS) loci, defined using only a single cut-off (PS = −0.71) (Supplementary Fig. 6b). While such a deconvolution likely oversimplifies the reality of pairing we nevertheless used it to bootstrap our investigation forward. Specifically, we divided the Hi-C amenable portion of the whole genome into regions of tight and loose pairing by first classifying each domain as either tightly or loosely paired based on its PS, and then merging consecutive domains of the same pairing type into one region (see "Methods"). According to this classification procedure, ~34% of the genome is loosely paired, and ~66% is tightly paired (Supplementary Fig. 6c, d).

**Tightly and loosely paired regions vary in organization**. To better understand chromosome organization within tightly and loosely paired regions, we selected those spanning distances large

enough for us to conduct our studies (200–400 kb or 100–200 kb; "Methods", Supplementary Fig. 7) and calculated $P_{thom}(s)$ and $P_{cis}(s)$. Tightly and loosely paired regions differed in the decay of *cis* and *thom* contact frequencies. Within the 200–400 kb tightly paired regions (Fig. 2h), *thom* contacts at the highest registration (smallest genomic separation, s = 1 kb) appeared as frequent as *cis* contacts at s = ~5 kb. In loose regions, the frequency of such *thom* contacts matched that of *cis* contacts at s = ~30 kb (Fig. 2h, marked on graph). This indicated that, in loose regions, homologs were aligned less precisely. Interestingly, we found that regions of tight and loose pairing also differed in their internal organization. This was evident from the different shapes of their $P_{cis}(s)$ curves —in tight regions, the $P_{cis}(s)$ curve had two modes (Fig. 2h, left), a shallow mode at s < ~30 kb and a steep mode at s > ~30 kb, while in loosely paired regions, we observed only a shallow mode (Fig. 2h, right). Drawing from other Hi-C studies, where the presence of a shallow mode followed by steep mode is a signature of domains[38,39], we then further interpreted our *cis* data. In particular, the transition of $P_{cis}(s)$ at ~10–30 kb within 200–400 kb tightly paired regions, suggested that they consisted of a series of relatively small domains, within which pairing may reflect primarily the constraints imposed by tight pairing at the boundaries. In contrast, we did not see a similar transition of $P_{cis}(s)$ within these 200–400 kb loosely paired regions, suggesting that each of these regions constituted a single domain. This distinction between tight and loose regions is also evident from visual inspection of the data (Fig. 2f).

The association of higher pairing scores with domain boundaries (Fig. 2g) was particularly intriguing, given that domain boundaries are enriched in insulator and architectural proteins (refs. [5,40] and reviewed by ref. [41]), the observation that some insulator proteins and insulator elements promote transvection (refs. [16,19,42,43], and reviewed by ref. [44]), and the enrichment of architectural proteins at genomic sites involved in allelic interactions[27]. Indeed, despite a few discrepancies among different published ChIP-seq datasets, many insulator proteins were enriched at PnM boundaries, with strong correlations between the ChIP-seq peaks and PS for some (e.g., Nup98, with the highest correlation coefficient) and a weak anti-correlation for others (e.g., Su(Hw), with the weakest correlation coefficient) (Supplementary Table 3). This analysis indicates a potential structural and functional regulatory role for some of these proteins on a genome-wide scale, such that, they would form insulated *thom* and *cis* domains and contribute to pairing.

**Pairing types associate differentially with gene expression.** Having elucidated the structure of paired homologs and its variation, we next addressed the question of whether homolog pairing may bear a genome-wide relationship to genome function. In particular, we conducted three genome-wide analyses, assessing whether pairing correlates with specific epigenetically defined types of chromatin, A- or B-type compartments, and/or gene expression. With respect to chromatin types, we turned to the five defined by Filion et al.[45], in *Drosophila*, wherein Polycomb group (PcG) repressed chromatin (H3K27me3 enriched) is dubbed blue, inactive chromatin (lacking epigenetic marks) is dubbed black, heterochromatin (HP1 associated and H3K9me3 enriched) is dubbed green, and active chromatin within enhancers/promoters (H3K36me3 depleted) and gene bodies (H3K36me3 enriched) are dubbed red and yellow, respectively. By comparing the coordinates for chromatin types identified in $Kc_{167}$ cells to the PS track, we found that even a localized survey of 1.5 megabase (Mb) of the genome revealed that low PS regions coincide with inactive (black) and repressed (blue) chromatin types, while active chromatin (yellow and red) is present in

regions of high PS (Fig. 3a). Excitingly, some of these trends were confirmed globally, with active regions enriched for high PS, heterochromatin (green) showing a bimodal distribution, and repressed (blue) and inactive (black) chromatin containing regions of both high and low PS (Fig. 3b).

Next, we examined the relationship between pairing and the 3D spatial compartmentalization of active and inactive chromatin[2]. Here, we observed a strong correlation between high PS values and the *cis* eigenvector track (a measure of compartments as determined from Hi-C maps) in individual genomic regions (Supplementary Fig. 8) as well as genome-wide (Spearman's correlation coefficient $(r_s) = 0.71$, $p < 10^{-10}$; Fig. 3c). Regions with high PS values and thus likely to be tightly paired were in predominantly active A-type compartments (54.4% of mappable genome) as versus inactive B-type compartments (12.6%). Conversely, regions with lower PS values and thus likely to be loosely paired were more often in B-type (25.2%) as versus A-type (7.9%) compartments (Fig. 3c). In short, homolog pairing was correlated with compartmentalization of the genome, and active A-type compartments were more likely to be tightly paired. As compartmentalization of the genome into active and inactive compartments may be independent of TAD formation in mammals[38], our observations may suggest that, pairing may be more related to compartments, gene expression, and the epigenetic states governing them. Taken together with the recent report on the major role of compartmentalization in *Drosophila cis* genome architecture[46], these observations put compartmentalization as the main force behind formation of both *cis* and *thom* genome architecture.

Finally, we performed RNA-seq to assess gene expression in PnM cells and found that pairing correlates with gene expression in individual genomic regions (Supplementary Fig. 8) as well as genome-wide ($r_s = 0.40$, $p < 10^{-10}$; Fig. 3d); regions that are expressed are predominantly tightly paired and have high values of PS (31.9% mappable genome), with only a small percentage (4.8%) of expressed loci being loosely paired (Fig. 3d). On the other hand, lack of expression is not predictive of the degree of pairing; lowly-expressed regions can be associated with either high or low PS values (36.1% and 27.3%, respectively). These analyses indicated that most active regions (A compartments or regions of high expression) are tightly paired, while repressed and inactive regions demonstrated a variable degree of pairing. In brief, all three approaches argue strongly that pairing bears a genome-wide relationship to genome function.

As many loci in *Drosophila* have been documented to support transvection, we categorized the commonly recognized autosomal loci with respect to whether they resided in tightly or loosely paired regions in the PnM genome. Fifteen of the seventeen loci for which transvection or related pairing-related phenomena have been reported are associated with tight pairing (with two also associated with loose pairing) and, thus, fall in the upper quadrants of Fig. 3d (Supplementary Table 4). Excitingly, this list includes the Antennapedia and Bithorax complexes (ANT-C and BX-C), which include HOX genes critical for body segmentation in *Drosophila*[47]. While these loci are known to interact[7,48], our data explicitly reveal their interaction in *thom* (Supplementary Fig. 9) and show their co-localization within the same compartment.

**Knock-down of pairing factors disrupts pairing globally.** Our final goal was to determine the potential of PnM cells to develop into a robust system for interrogating the mechanism of pairing. We aimed to determine, first, whether PnM cells are responsive to dsRNA, second, whether knockdown of genes involved in pairing[11,28–30,49–51] would affect pairing and, third, whether

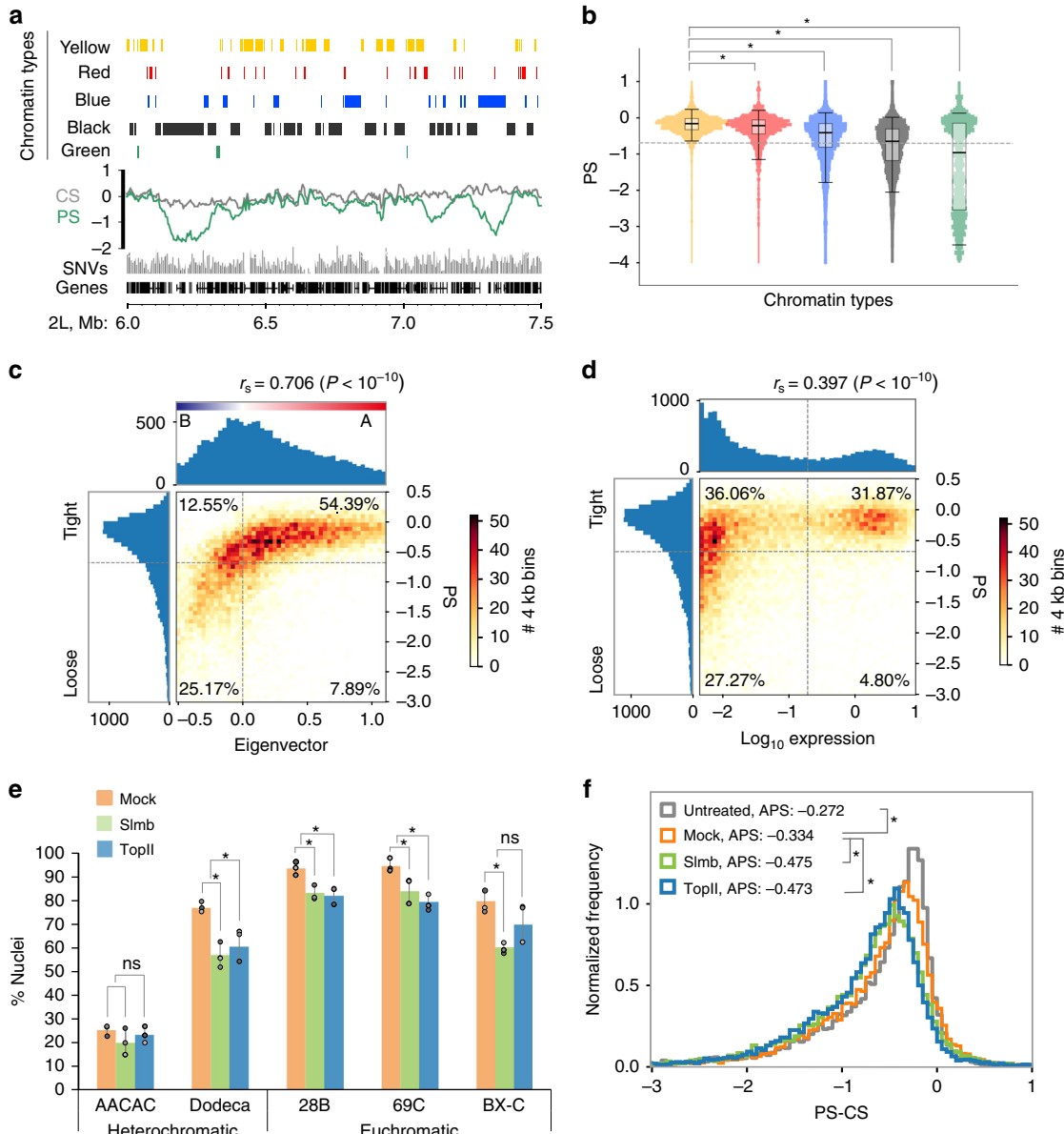

**Fig. 3** Pairing in PnM cells is correlated with active genomic regions and can be disrupted by RNAi. **a** 1.5 Mb region of 2 L: Pairing scores (PS) and *cis* scores (CS) shown in a genome browser as compared to chromatin types identified in $Kc_{167}$ cells[45] and with SNVs in PnM cells. **b** Normalized distributions of PS within regions of different chromatin types. Dashed line shows the threshold between tight and loose pairing. *$p < 1^{-10}$, Mood's test against yellow chromatin. **c** Distribution of PS values relative to the eigenvector shows that pairing is correlated with compartmentalization, A-type compartments being almost exclusively tightly paired, Spearman's correlation coefficient ($r_s$) = 0.706, $p < 10^{-10}$. **d** Distribution of PS values relative to gene expression in PnM cells shows that expressed genes are almost exclusively tightly paired, Spearman's correlation coefficient ($r_s$) = 0.397, $p < 10^{-10}$. **e** Levels of pairing (quantified and displayed as in Fig. 1d, in PnM cells after Slmb and TopII knockdown showed a reduction as compared to the control (mock) at all loci (*$P < = 0.05$, unpaired *t*-test) except for the AACAC satellite repeat and, in the case of TopII knockdown, at BX-C (ns, non-significant; error bars, s.d for three biological replicates; for $N > 100$ nuclei/replicate). Source data are provided as a Source Data file. **f** After Slmb and TopII knockdown, the aggregated pairing score (APS) values were reduced by 0.203% and 0.201%, respectively, as compared to untreated sample ($p < 0.001$) and 0.141% and 0.139%, respectively, as compared to mock ($p < 0.001$). The 0.062% reduction in mock as compared to untreated samples was also significant ($p < 0.001$). *p*-values determined using bootstrapping (Methods)

disruptions of pairing as detected by FISH would be detectable via Hi-C. These issues were key. While previous studies had successfully disrupted pairing in *Drosophila* cell lines, under no circumstance had the pairing in diploid cells been disrupted beyond ~50%[29,50] due probably at least to the incomplete nature of RNAi-directed knockdown and perdurance of gene products. It is also possible that pairing, once established, is not easily disrupted[37,50]. Finally, disruptions of pairing might affect primarily those forms that are not amenable to detection by Hi-C; as FISH

studies often consider two loci to be paired when the center to center distances of the corresponding FISH signals are as far apart 0.5 to 1.0 μm, it is possible that some changes in pairing cannot be captured by Hi-C.

Excitingly, using RNAi to target two genes known to promote pairing, Slmb (component of $SCF^{Slmb}$ complex[28–30]) and Topoisomerase II ((TopII[50]), we reduced the corresponding mRNA levels in PnM cells by 75.2 ± 2.8% and 82.5 ± 5.0%, respectively (Supplementary Fig. 10a). Importantly, we observed a

concomitant 14.1–18.5% reduction of pairing as assayed by FISH (Fig. 3e; Supplementary Table 5). While this reduction was less than previously reported[29,50], it was significant as compared to mock RNAi trials ($p < 0.05$, unpaired $t$-test) (Fig. 3e). We also attempted a stronger disruption of pairing by overexpressing Cap-H2[28,52], but found that levels of pairing were not affected significantly at those loci (Supplementary Fig. 11). Since knockdown experiments were more disruptive of pairing, we generated Hi-C maps for the knockdown and mock experiments, each with about 20 million haplotyped mappable reads (Supplementary Table 2) and found a reduction in $P_{thom}(s)/P_{cis}(s)$ for both Slmb and TopII RNAi samples at all separations as compared to mock and untreated sample (Supplementary Fig. 10b; error bars for each sample fall within lines). Note that, the values of $P_{thom}(s)/P_{cis}(s)$ for the mock samples veer below those for untreated controls at genomic separations greater than 100 kb. While this reduction suggests that the knockdown treatment may perturb *thom* and/or *cis* interactions and thus may be interesting in and of itself, our focus has been on the even greater reduction in $P_{thom}(s)/P_{cis}(s)$ for both RNAi-treated samples (Supplementary Fig. 10b). Slmb and TopII knockdowns also produced a change in PS. To quantify this change, we computed the aggregated pairing score (APS) as the mode of (PS-CS) distribution, which summarizes the degree of pairing with a single value ("Methods"). As shown in Fig. 3f, APS dropped after knockdown of Slmb or TopII, as compared to mock, and the untreated sample. These observations were consistent across replicates (Supplementary Fig. 10c) and across tight and loose regions (Supplementary Fig. 12; "Methods"). In addition, some *thom* interaction peaks in Slmb and TopII RNAi samples were depleted as compared to Mock sample (Supplementary Fig. 13). In summary, not only were PnM cells amenable to RNAi, but Hi-C could detect global changes in pairing as a result of the knockdown of pairing factors.

## Discussion

In conclusion, we established a hybrid, fully phased PnM cell line, which allowed us to use haplotype-resolved Hi-C to distinguish *cis* and *thom* interactions, revealing great detail in the structure of homolog pairing (Fig. 4a, b), in addition to uncovering a genome-wide correlation between pairing and gene expression. Using SNVs and our haplotype-resolved approach, we find that pairing is extensive, spanning a wide range of genomic distances, from as small as few kilobases to as large as tens of megabases, and includes *thom* domains, loops, and compartments. Furthermore, we observed two forms of pairing (Fig. 4b): a tighter, more precise form that can encompass many contiguous small domains paired at their boundaries and a looser, less precise form often corresponding to single domains flanked by tight pairing at the boundaries. The relationship between loose pairing and domains is consistent with both a transgene-based study[53] as well as super-resolution images of juxtaposed domains[23].

We also examined the relationship between pairing and genome function, discovering that tight pairing can be correlated with either expressed or repressed regions, while loose pairing is correlated primarily with repression or inactive chromatin. While this finding may suggest that gene activity can, but does not always, promote tight pairing, it is also possible that tight pairing facilitates the formation of microenvironments that can, but do not always, favor transcription. Such microenvironments may promote the entangling of R-loops (Fig. 4bi) or enrichment of RNA polymerase, transcription factors[21,54,55], and/or insulator elements and associated proteins at domain boundaries[4,5,21,27,56,57]. In brief, a pairing-mediated microenvironment may result in a more robust level of either expression or repression (Fig. 4bi, ii) (Supplementary Table 4), consistent with

the association of transvection with both gene activation and gene repression (reviewed by refs. [8–12]). It is also possible that the different types of pairing promote or antagonize allele-specific expression; further investigation of these scenarios will require single-cell analyses. Note that our findings differ from predictions of a study[20] that, in the absence of haplotype-resolved data, was nevertheless able to simulate pairing via the computational integration of Hi-C and lamina-DamID data representing embryos[7] and $Kc_{167}$ cells[58], respectively. Contrary to our findings, the simulations predicted correlations between active regions and loose or tight pairing, and between inactive regions and tight pairing. One possible explanation is that these two studies suggest an as yet unexplored mechanism of pairing.

Loosely paired regions are equally interesting. First, unlike tightly paired domains, which are associated with both active as well as repressed regions, loosely paired domains show a preference for repressed genomic regions, with just a small percentage of the genome being both loosely paired and expressed (Fig. 3c). While these observations may suggest that inactive genomic regions lead to loose pairing, it is also possible that loose pairing is inherently not permissive of transcription. If the latter were true, and speculating broadly, then it may be that achieving tight pairing could be a first step in becoming susceptible to regulation at some loci.

Loosely paired regions are also interesting because they lack a *thom* diagonal, indicating lack of railroad track pairing within some *thom* domains (Fig. 2f, schematics below). Importantly, the boundaries of these loosely paired regions are tightly paired, supporting a model that integrates pairing, loop formation, and chromosome compaction via a mechanism wherein chromosomes are looped (buckled out) by anti-pairing between regions of pairing[29]. In these loosely paired regions, homologs (and, perhaps, sister chromatids) could be extruded or formed via some other mechanism[39] and/or anti-paired between tightly paired regions but still interact by virtue of remaining tightly paired at their loop bases. In this scenario, tightly paired regions could behave as extrusion barriers and become boundaries (Fig. 4c). Interestingly, RNA polymerase and insulator proteins have been proposed to behave as barriers to extrusion and thus may be in play in this scenario (ref. [59]; Supplementary Table 3). Lack of a diagonal has also been observed for polytenized chromosomes[60,61], where it may reflect an outnumbering of *cis* contacts by an abundance of *trans* contacts. These observations raise the possibility that, under some circumstances, there may be competitive relationships between short-range *cis* and *thom* contacts[11,62] and/or between short-range and long-range *thom* contacts.

Finally, we turn to our observation that loci interacted with a second locus in *thom* nearly as often as they did in *cis*. Researchers have long speculated about the consequences of providing regulatory regions with a *cis-trans* choice (reviewed by refs. [9,11]), wherein pairing could enhance the co-regulation of allelic regions or enable transcriptional states to be transferred from one chromosome to another[14,63,64]. Indeed, as interactions in *cis* may preclude interactions in *trans*, the *cis-trans* choice recalls a hypothesis wherein regulation of a genomic region may require a balance, perhaps even a dynamic interplay, between pairing and unpairing, such as in a model counterbalancing pairing (linear locking)[62] with unpairing (looping or buckling out) (Fig. 4c). In other words, finely regulated genomic regions may need to be poised to pair or unpair on a moment's notice. A capacity to transition easily between different states of pairing and unpairing may even promote or antagonize allele-specific expression, where the unpaired state may facilitate allele-specific expression, especially in mammals (reviewed by refs. [9,11]). Homolog pairing may even accomplish what compartments do in

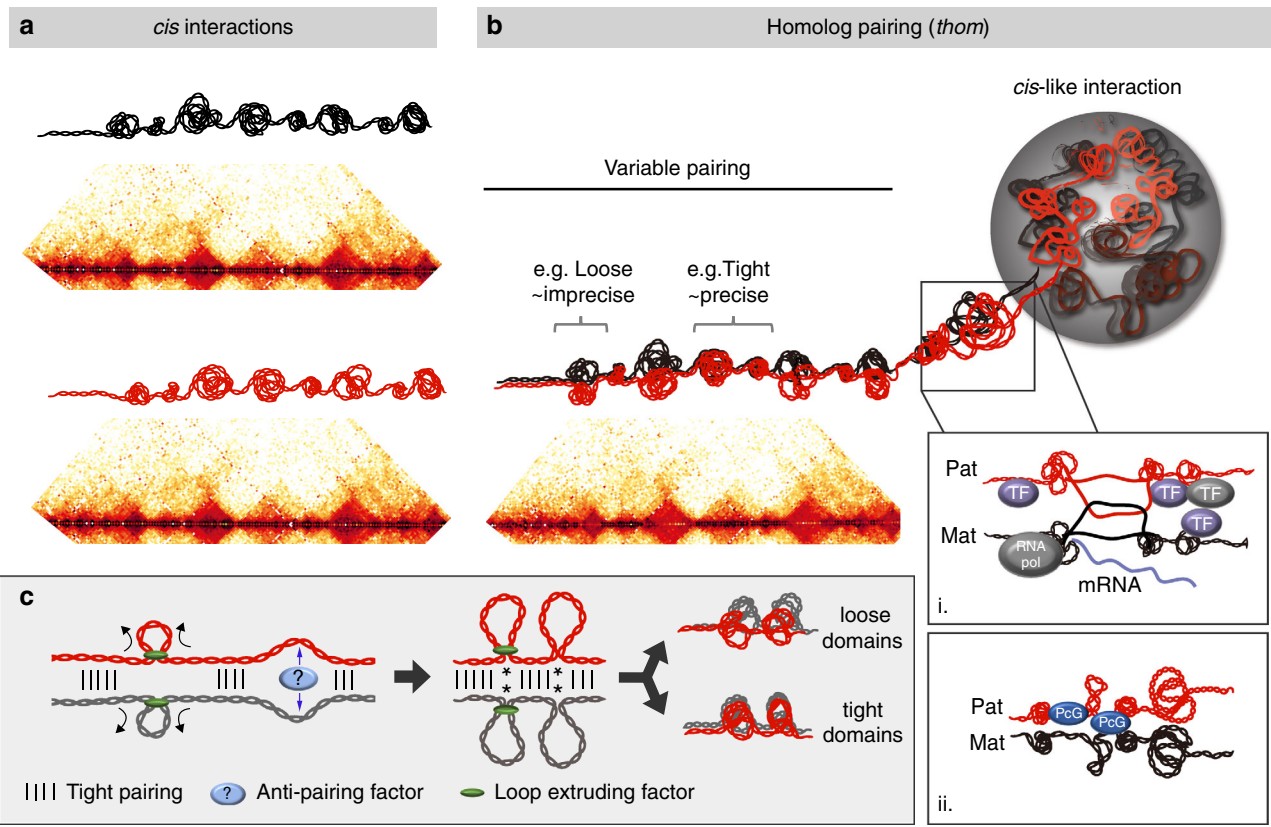

**Fig. 4** Haplotype-resolved Hi-C in PnM cells distinguished different forms of *thom* from *cis* interactions. **a** *Cis* contact maps for two homologs and schematics depicting possible *cis* interactions. **b** *Thom* contact map demonstrating a variable structure of pairing, including tight, precise pairing interspersed with looser, less precise pairing. *Thom* interactions encompass organizational structures that are concordant with *cis* behavior and could facilitate a variety of transcriptional states, including i. active or ii. repressive environments. **c** Left: Homologous loops between tightly paired regions may form by extrusion (black arrows), anti-pairing (blue arrows), or a combination of both. Right: Here, loops could result in *thom* domains that are either loosely paired (top), or railroad-track paired throughout if they fold back on each other, (bottom), with *cis*-maternal and *cis*-paternal domain boundaries concordant in both scenarios. Note, loop extrusion in mammalian systems is proposed to involve a cohesin ring[79,80] (green oval) through which a single chromosome passes, suggesting that extrusion in the context of pairing may involve the passage of each homolog separately (shown) or two homologs simultaneously (not shown), the outcome of which could be either loosely or tightly paired domains. In a nonexclusive alternative, loops are formed by anti-pairing[29], where pairing might be counterbalanced with unpairing via anti-pairing factors such as Cap-H2. Interestingly, extrusion and/or anti-pairing could bring enhancers and promoters together at the base of the loops (indicated by *), activating transcription, such as might happen in tightly paired regions. In the context of anti-pairing, this could explain the curious co-localization of Cap-H2 with regions of tight pairing (Supplementary Table 3; also ref. [27])

both *cis* and *trans*, and what domains do in *cis*, co-localizing genomic regions to achieve an "economy of control[64]". How structurally and functionally independent these processes are, remains to be explored.

## Methods

**Fly stocks and crosses**. Highly divergent parental fly lines from the *Drosophila* Genetic Reference Panel were selected[65], and a cross was set up between DGRP-057 virgin females and DGRP-439 males[33]. Primary cultures were established from an overnight embryo collection aged for 2 h (2–14 h AEL).

**Establishing primary culture PnM and clonal cell line**. Primary cell line was generated as described previously[66]. Embryos produced from a cross between divergent parents, about 2–14 h after egg lay, were collected overnight at 25 °C on agar juice plates covered with killed yeast paste. After the embryos were rinsed from the plates and collected in a 50 ml sieve basket they were washed using TXN wash buffer (0.7% NaCl, 0.02% Triton X-100). The TXN was replaced with 50% bleach in water for 5 min to dechorionate and surface sterilize the embryos. The embryos were washed extensively with TXN, sprayed with 70% ethanol, and transferred to a sterile tissue culture hood, where they were rinsed once in water and placed in 3 ml medium for 5 min (unsupplemented M3 Insect Media (Sigma)). Around 100 μl embryos or more were transferred to a homogenizer (Wheaton 5 ml) and homogenized in 5 ml M3 Insect medium with a loose pestle 3/4 of the way until embryos were homogenized, then 5× with a loose pestle all the way, followed by 5× with a tight pestle. The homogenate was transferred to a 15 ml conical tube,

and cell debris was pelleted at 500 rpm. The supernatant was transferred to a new tube and cells were pelleted by centrifugation at 1200 rpm, resuspended in fresh media and counted. About 8 million cells were plated in 25 cm² T-flasks and grown at 22 °C with 5 ml of supplemented M3 media (M3 Insect Media, 10% Fetal Bovine Serum (JRH), 2.5% Fly Extract (DGRC), 0.5 mg ml⁻¹ Insulin (Sigma), 1:100 Pen-Strep (Gibco)). After a couple of weeks, the media was replaced with conditioned supplemented M3 media (0.2 μm filter-sterilized used media) and was then changed every couple of weeks for a few months. With continued passaging the cells became stable and immortalized spontaneously. Then, as they divided more regularly, they were trypsinized with TrypLE 1 × (ThermoSci) and split at 1:5 into supplemented M3 media every 2–3 days. Establishing a clonal cell line was done using a serial dilution single cell cloning protocol http://www.level.com.tw/html/ezcatfiles/vipweb20/img/img/34963/3-2Single_cell_cloning_protocol.pdf. One of the clonal lines isolated, Pat and Mat line #A1 (PnMA1), is used in this study and is referred to as PnM.

**Cell culture**. Kc₁₆₇ (stock #1), and clone8 (cl.8+) (stock #151) cells were obtained from the Drosophila Genome Resource Center and cultured according to standard protocols at 22 °C (see www.flyrnai.org for more details). Hybrid PnM cells were cultured as cl.8+ cells with supplemented M3 media (supplemented M3 media (M3 Insect Media, 10% Fetal Bovine Serum (JRH), 2.5% Fly Extract (DGRC), 0.5 mg ml⁻¹ Insulin (Sigma), 1:100 Pen-Strep (Gibco)), according to standard protocols at 25 °C in small culture dishes (100 × 20 mm), with the following modifications: the cells were trypsinized and split at 1:5, or 1:10 every other day, and the media was replaced every day after they are washed with 1× PBS. To ensure the cells were in log phase for any experiment, they were split 12–18 h earlier.

**Genomic DNA extraction and PnM genome sequencing**. One to two million PnM cells were used to isolate Genomic DNA using Qiagen DNeasy Blood & Tissue Kit, and ~0.5 μg genomic DNA was used for the generation of an Illumina TruSeq Nano DNA Library. The library was 150 base pair paired-end sequenced using Illumina HiSeq2500 at TUCF Genomics core.

**FACS**. Cells were split a day prior and then 1–3 million cells were harvested and fixed with 95% ethanol, washed with 1× PBS and stained at a concentration of 30 μM FxCycle PI/RNase staining solution (Life Technologies) for 30 min at room temperature. Cell populations were assayed based on DNA content to determine their cell cycle profile using an LSR II Analyzer at the HMS Immunology flow cytometry core.

**Preparing metaphase spreads and karyotyping**. Metaphase cells were prepared using protocols adapted from published methods[67]. Hundred and fifty microlitersof colchicine was added to 5 ml cells in culture at a concentration of 30 μM for 45 min prior to fixation and spread preparation. Cells were spun down at 1200 rpm, washed with 1× PBS and resuspended in 10 ml 1% sodium citrate slowly, while vortexing regularly. The cells were incubated at room temperature for 30 min. Then 1 ml of cold fixative was added (3:1 methanol: glacial acetic acid solution) while vortexing gently, cells were spun down, and washed three more times in 10 ml of the same fixative. Finally, cells were resuspended in 1 ml of the fixative and dropped at a height of 5 inches or more onto a glass slide under humidified conditions. The slide was allowed to dry in a humidified chamber and then washed in 70, 90, and 100% ethanol successively. For long-term used they were stored at 4 °C, in 1× PBS. In order to examine the karyotype, slides were mounted with Slowfade Gold Antifade with DAPI (Invitrogen). The spreads were then examined with a Nikon Eclipse Ti at 60×. The karyotype was examined for PnM cells for $N = 50$ from two different metaphase spread preparations, and showed the cell line to be male and diploid, apart from chromosome four. The fourth chromosome was 7% diploid, since it was either monosomic (~71%) or trisomic (~22%). Given its ploidy, and that it is largely heterochromatic, with an average 1 SNV per kb (compared 5 SNVs per kb for the second or third, Supplementary Table 1), it was not included in the haplotype-resolved Hi-C mapping and analysis. Karyotyping, in combination with homolog-specific FISH proved PnM cells to be male, diploid, and hybrid.

**FISH probes**. Heterochromatic repeat regions were assayed using previously described FISH probe sequences, and used to assay FISH loci localization as shown previously[29,50]. The probes used were synthesized by Integrated DNA Technologies (IDT) as follows: Alexa647-359-X: /5Alex647N/-GGGATCGTTAGCACTGGTAATTAGCTGC, Atto565-AACAC-II: /5Atto565N/ AACACAACACAACACAACACAACACAACACAACAC, and Alexa488-dodeca-III: /5Alex488N/-ACGGGACCAGTACGG. The euchromatic Oligopaints FISH probes used in this study, 16E, 69C and 28B as well as HOPs probes were described previously[50,68,69] and were generated using the T7 method[70]. The Oligopaint libraries used as template DNA are described in Supplementary Table 6 and are amplified using the primers shown in Supplementary Table 7. Secondary probe binding sites were added to Oigopaints' mainstreets and T7 sequences to backstreets by touch-up PCR. For detecting the euchromatic Oligopaints, a secondary fluor-tagged probe sequence complementary to the mainstreet, was co-hybridized with the primary probe. Homolog-specific Oligopaints (HOPs) probes include at least one SNV location per Oligopaint and are used to distinguish either 2L or 3L homologs in the hybrid line by targetting a 2 Mb sub-telomeric region on either chromosome 2, and chromosome 3. The secondary, dual labeled probes used for detection of euchromatic targets are ordered from IDT and are as follows: Secondary1: /5Alex488N/CACACGCTCTTCCGTTCTATGCGACGTCGGTGagatgttt/3AlexF488N/, secondary5: /5Atto565N/ACACCCTTGCACGTCGTGGACCTCCTGCGCTA/3Atto565N/, and secondary6: /5Alex647N/TGATCGACCACGGCCAAGACGGAGAGCGTGTGagatgttt/3AlexF647N/.

**Metaphase FISH**. Metaphase FISH was done as described previously[67]. Slides from the metaphase spread preparation were rehydrated in 1× PBS for 5 min and denatured in 67% formamide/2× SSCT at 80 °C for 90 s, followed by washes in ice-cold 70, 90 and 100% ethanol. 20 pmol of HOPs were co-hybridized with 40 pmol of secondaries without any additional denaturation at 42 °C overnight. The remainder of the protocol is the same as the interphase FISH protocol.

**Interphase FISH**. Fluorescence in situ hybridization was done as in described previously[70]. A cell suspension of 0.5–1 million cells per ml was allowed to settle on poly-l-lysine coated slides for a few hours, washed with 1× PBS, then fixed in 4% paraforlamdehyde and washed in 1× PBS again. The slides were either used for FISH immediately or stored in 1× PBS at 4 °C. Just before FISH, the cells were permeabilized by incubating in 0.5% PBST for 15 min, and 10 min in 0.1 M HCl. The slides are then washed in 2× SSCT for 5 min (0.3 M sodium chloride, 0.03 M sodium citrate, 0.1% Tween-20), and (50% formamide, 2× SSCT). FISH slides were incubated in (50% formamide, 2× SSCT) 60 °C for 20 min. FISH probes were added in a hybridization solution of (10% dextran sulfate, 2× SSCT, 50% formamide) and 100 pmol of heterochromatic, or 50 pmol euchromatic probes, 16E, 28B, 69C and BX-C per hybridization. The slides were then denatured by placing them on a heat block at 80 °C for 3 min and allowed to co-hybridize overnight at 42 °C with 40 pmol of secondary probe. Following hybridization, slides were washed in 2× SSCT at 60 °C for 20 min, 2× SSCT at room temperature for 5 min, and 0.2× SSC at room temperature for 10 min before being mounted using Slowfade Gold Antifade with DAPI (Invitrogen) and imaged. To quantify level of pairing for the second and third chromosomes at the same time, probes were co-hybridized for 28B-II, 69C-III FISH probes or AACAC-II, and dodeca-III. Since the cell line is male, to score cells that are non-replicating, we used probes targeting the X chromosomes in our co-hybridization reactions, using either euchromatic target 16E or heterochromatic target, 359, and consider the pairing only in cells with one X-specific FISH signal per nuclei.

**Image acquisition and analysis**. All images were obtained using Nikon Eclipse Ti microscope with a 60X oil objective and Nikon ND acquisition software. The raw TIFF files obtained were analyzed using custom-written MATLAB scripts[29] and later adapted[67] for measuring different properties such as the number of FISH dots per nucleus. All uniquely identifiable foci of fluorescent signal (above background) were counted as FISH signals. The number of FISH foci were also counted manually to confirm consistency and determine the degree of localization of FISH foci in 3D. Homologous loci were considered paired if FISH signals targeting the loci co-localized (i.e., gave a single signal) or exhibited a center-to-center distance of ≤0.8 μm.

**Immunostaining to determine cell type and mitotic index**. PnM cells were fixed with 4% formaldehyde for 20 min, following previously published protocols[67]. In order to determine the mitotic index for the PnM hybrid clone, a primary antibody against phosphohistone H3 (P-H3; rabbit used at 1:100; Epitomics) was used for immunofluorescence in a 1× PBS buffer. A Cy3-conjugated anti-rabbit secondary antibody (Jackson ImmunoResearch Laboratories) was used at 1:100. Mitotic index for PnM cells was determined to be 2.48% ± 0.78, $N = 1300$.

In order to determine cell type as described previously[71], PnM were immunostained using rabbit anti-dMef2 antibody, which was a gift from Bruce Paterson[72]. The antibody was used at 1:1000, with a Cy3-conjugated anti-rabbit secondary antibody at 1:100 (Jackson immunoresearch). The cells expressed dMef2 exclusively, and are most likely of mesodermal origin, as they tested negative for other cell type markers including an epithelial cell marker; D-E-Cadherin, ((Rat)-anti E-Cadherin 1:5 (Hybridoma Bank, Iowa), a fat cell marker; Nile red solution, (Sigma; 1% stock in DMSO diluted to 1:5000), and a nerve cell marker; horse Radish peroxidase (HRP) (Jackson immunoresearch (Rhodamine conjugated) 1:200).

**dsRNA synthesis and RNAi treatment**. Synthesis of dsRNA was carried out according to standard protocols (see www.flyrnai.org for more details). Control cells were treated with blank dionized water. Primers used for dsRNA synthesis are listed in Supplementary Table 8. The dsRNA was administered to the cells using a calcium phosphate transfection kit (Invitrogen) in 60-mm well plates. Kc167 cells were seeded at 2 million cells per ml and treated with 15 μg dsRNA and harvested after 3 days. PnM cells were seeded at 1 million cells per ml and treated with 30 μg dsRNA on the first and third day, and then harvested on the fourth. Cells from the knockdowns were counted and an aliquot was taken for RNA isolation, and once knockdown of mRNA was confirmed cells were split to be fixed for Hi-C, and for FISH. Two biological replicates were processed for each treatment. Knockdowns in Kc167 cells were used as a control for the quantification of knocking down mRNA but were not processed for Hi-C experiments.

**Cap-H2 overexpression**. Wild-type Cap-H2 was gateway cloned into Actin-driven, venus-tagged vector (Avw-Cap-H2). Cells were transiently transfected using a calcium phosphate transfection kit (Invitrogen) in 60-mm well (at 1 million cells per ml), according to manufacturing recommendations, apart from adding a booster transfection at day 4 and harvesting cells at day 6. Cells were fixed and prepared for FISH as described previously.

**qPCR**. Quantitative PCR was used to assay efficiency of RNAi knockdowns according to standard techniques. Total RNA was isolated from cells using a Qiagen RNeasy Plus kit and then converted to cDNA using SuperScript VILO cDNA synthesis kit (Invitrogen) for RT-PCR. Primers for qPCR were designed using Primer3 website (http://bioinfo.ut.ee/primer3-0.4.0/) and are listed in Supplementary Table 9. Reactions were set up according to recommended protocol using iQ SYBR Green Supermix (BioRad) and run on BioRad CFX Connect Real-Time System at an annealing temperature of 58 °C. BioRad software determined CT values for qPCR reactions, and the level of knockdown is determined using the $2^{(-\Delta\Delta CT)}$ method[73]. The level of knockdown was determined for two biological replicates and normalized to two controls; Act5c, and RP49. Cells that shows a significant drop in mRNA levels were later processed for Hi-C experiments.

**In situ Hi-C protocol**. This protocol was adapted from a previously published protocol[32] with modifications. Unless otherwise specified, 75 T flasks of PnM cells were cultured to 70% confluency, then washed with FBS-free Schneider's medium,

and crosslinked with 1% formaldehyde for 10 min at room temperature. Fixation was quenched with 1 M glycine solution, and the cells were scraped gently off the flask, spun down at 1200 rpm, and washed once more with 1× PBS. Supernatant was removed, and cells were resuspended in 1× PBS and then ~2.5 million cells were counted to be used for the rest of the protocol. Nuclei were permeabilized with ice cold lysis buffer (10 mM Tris-HCl pH 8.0, 10 mM NaCl, 0. 2% Igepal supplemented with 5X Complete, EDTA-free Protease inhibitors (Roche). DNA was digested with 500 units of DpnII, and the ends of restriction fragments were labeled using biotinylated nucleotides and ligated in a small volume. After reversal of crosslinks, ligated DNA was purified and sheared to a length of ~700 base pairs with QSonica sonicator (30% power, 30 s on, 30 off for 20 min, at which point ligation junctions were pulled down with Dynabeads MyOne Streptavidin beads (Invitrogen) and prepped for Illumina sequencing. Aliquots at different steps were taken to measure the concentration with Qubit dsDNA HS assay kit and run on a 2% agarose gel with SYBR Gold nucleic acid gel stain (1:10000) to ensure quality of the sample prepared, and efficient digest, ligation, and binding to the beads. For library preparation, we used BioNEXTFLEX barcode-6 (Bioo Scientific) and followed manufacturer recommendations, amplified our final library with Q5 HIFI Hot Start High Fidelity PCR and Bioscientific primer mix for six cycles on the beads. The final product was then diluted to 250 μl with 1 mM Tris-HCl, and separated on a magnet, then transferred the supernatant to a new tube and purified with 0.7× AMPure XP bead (Beckman Coulter). Incubation times are extended to 15 min to maximize library recovery. To remove traces of short products, we resuspend beads in 100 μl of 1× Tris buffer and add another 70 μl of AMPure XP beads. Mix by pipetting 20× and incubated at room temperature for 15 min, and separated for another 15 min, and continue with the washes and drying as previously described. Once the beads are dry, we resuspended in 20 μL 1× Tris-HCl, incubated for 15 min, then separated for another 15 min on the magnet and transferred the final library to a fresh tube. Two replicates were prepared per sample. The library quality was assessed using the High Sensitivity DNA assay on a 2100 Bioanalyzer system (Agilent Technologies). Then were 150 base pair paired-end sequenced using Illumina HiSeq2500 at TUCF Genomics core. PnM untreated samples (two replicates) were sequenced in four lanes, while each of the RNAi treatments (two replicates) was sequenced in one lane.

**RNA isolation and library preparation**. Total PnM RNA was purified from ~8 million cells per replicate using TRIzol (Life Technologies), followed by chloroform extraction, DNase treatment with RNase free DNase I recombinant (Roche), and clean up with RNeasy Mini kit (Qiagen). The quality of total RNA was determined using Agilent RNA 6000 Pico assay on a 2100 Bioanalyzer system (Agilent Technologies). RNA-Seq libraries were prepared using NEBNext Poly(A) mRNA Magnetic Isolation Module and NEBNext Ultra Directional RNA Library Prep Kit for Illumina according to the manufacturer's instructions. Poly(A) + RNA was enriched from 1 μg of total RNA, fragmented for 10 min at 94 °C, and reverse transcribed in the first strand cDNA synthesis with random primers. After adaptor ligation, cDNA was size selected between 400–600 base pairs with Agencourt AMPure XP beads, and amplified for 12 PCR cycles. The library quality was assessed using the High Sensitivity DNA assay on a 2100 Bioanalyzer system (Agilent Technologies). RNA-Seq libraries corresponding to three biological replicates were 150 base pair paired-end sequenced with Illumina HiSeq2500 at TUCF Genomics core.

**Western blot**. Whole cell extracts were prepared after 6 days of transient transfections and their protein levels were analyzed according to standard protocols. Blots were probed using a rabbit anti-GFP antibody (ab290, 1:1000) to detect levels of Venus-Cap-H2-WT, and a mouse HRP-conjugated anti-α-tubulin antibody (ab40742; 1:5000) was used as a loading control. Probing with anti-GFP was followed by a rabbit secondary antibody conjugated to HRP (at 1:5000:). Blots were stained using Pierce ECL Western Blotting Substrate (ThermoFisher Scientific).

**The construction of diploid PnM fly genome**. We generated the diploid genome (hybrid PnM genome) using (i) the homozygous autosomal PnM SNVs, (ii) the heterozygous phased autosomal PnM SNVs and (iii) the homozygous chrX PnM SNVs.

After sequencing the F1 PnM cell line at the average coverage of 396 reads per base pair [https://www.ncbi.nlm.nih.gov/pubmed/29096012], we used *bcftools* to detect the sequence variation of this library. We obtained high-quality normalized sequence variants using the following:

1. 'seqtk trimfq' to trim low-quality sequences, BWA mem, to align whole genome paired-end reads against the reference dm3 genome, and 'samtools–rmdup' to remove aligned PCR duplicates
2. 'bcftools pileup–min-MQ 20–min-BQ 20' to pile alignments along the reference genome.
3. 'bcftools call' to call raw sequence variants from the pileups.
4. 'bcftools norm' to normalize raw sequence variants.
5. 'bcftools filter INFO/DP > 80 & QUAL > 200 & (TYPE = "SNV" | IDV > 1)' to select only high-coverage high-quality normalized sequence variants using.

The two inbred parental fly lines (057 and 439) were sequenced at the average coverage of 118, and 117 reads per base pair, respectively, and treated similarly to detect sequence variation in their libraries as described in our companion paper[33]. Using the variants from PnM, and the two paternal lines, we then phased heterozygous PnM variants using 'bcftools isec'. Then, we picked high-confidence variants on the maternal autosomes by selecting heterozygous PnM variants that were present among maternal 057 variants and absent among paternal 439 sequence variants (both homo- and heterozygous); the high-confidence paternal variants phasing was selected in an opposite manner. Since PnM is a male line, for chrX, we considered only homozygous, high-quality variants detected in the PnM cell line. To reconstruct the consensus sequence of the paternal copy of chrX, we kept only homozygous variants detected in the paternal 439 fly line. Finally, we reconstructed the sequence of the PnM cell line with 'samtools consensus', using (a) the homozygous autosomal PnM SNVs, (b) the heterozygous phased autosomal PnM SNVs and (iii) the homozygous chrX PnM SNVs.

The number of reads from the hybrid genome, and parental lines used for phasing are summarized in Supplementary Table 10. Overall, WGS confirmed PnM cell line hybrid status, apart from uncovering a 24.6 Mb partial uniparental disomy of the right arm of chromosome 3 (chr3R). We adjusted our downstream analyses to take that into consideration (described in more detail later).

**Mapping and parsing**. Using the standard mode of *seqtk trimfq* v.1.2-r94 [https://github.com/lh3/seqtk], we trimmed low-quality base pairs at both ends of each side of sequenced of Hi-C molecules. This was followed by mapping the trimmed sequences to either, the reference dm3 genome, or the newly constructed dm3-based diploid genome (hybrid PnM genome) using *bwa mem* v.0.7.15 [https://arxiv.org/abs/1303.3997] with flags -SP.

We then used the *pairtools parse* command line tool (https://github.com/mirnylab/pairtools), to extract the coordinates of Hi-C contacts, kept read pairs that mapped uniquely to one of the two homologs, and used the standard mode of the *pairtools dedup* command line tool to remove PCR duplicates. Breakdown of the total number of reads recovered after the mapping and filtering process are summarized in Supplementary Table 2, and a summary of *cis* and *thom* reads recovered at different genomic separations is found in Supplementary Table 11.

Detailed methods estimating the percent of homolog misassignment was discussed in our companion paper[33].

**Contact probability P(s) curves**. To calculate the functions of contact frequency P(s) against genomic separation (s), we used unique Hi-C pairs, and grouped genomic distances between (10 base pairs and 10 megabases) into ranges of exponentially increasing widths, with eight ranges per order of magnitude. We found the number of observed *cis* or *trans*-homolog (*thom*) interactions, within this range of separations, and divided it by the total number of all loci pairs separated by such distances.

**Pairing score**. We introduced a genome-wide track called pairing score (PS), to measure the strength of the diagonal and to characterize the degree of pairing between homologous loci across the whole genome. The PS of a genomic bin is $\log_2$ of average *trans*-homolog (*thom*) Iteratively Corrected (IC) contact frequency (CF) between all pairs of bins within a window of +-W bins. For each genomic bin i, its PS with window size W is defined as:

$$\mathrm{PS}^W(i) = \log_2 \langle \mathrm{CF}_{m,n} \rangle,$$ averaged over bins $m$ and $n$ between $i-W$-th and $i+W$-th genomic bins on different homologs of the same chromosome.

As a control measure, we complemented the PS with a *cis* score (CS), which is calculated the same way as the PS, but over the *cis* contact map. Comparing the PS and CS value in a given bin reveals if the local variability in the PS is due to a change in homologous pairing in 3D (which affects the PS but does not affect the CS) or due to a theoretically possible local deviation from the equal visibility assumption of IC (which would affect both PS and CS equally). Using this definition, the PS quantifies only contacts between homologous loci and their close neighbors and not non-homologous loci on homologous chromosomes. We chose the window size W to be a balance between specificity and sensitivity. Increasing the window size increases sensitivity, accumulating contacts across more loci pairs, while smaller windows favors specificity, allowing to see smaller-scale variation of homologous pairing. For our contact maps with a 4 kb resolution, using a 7 × 7 bin window (W = 3) provided a balance between specificity and sensitivity. We interpreted the obtained PS tracks using a simple assumption: if a genomic locus was paired with its counterpart on the homologous chromosome in 100% of cells, the frequency of the *thom* contacts (i.e., PS) around the locus should be equal to the frequency of corresponding short-distance *cis* contacts (i.e., CS).

**Insulation scores**. Using the package *cooltools diamond_insulation* (https://github.com/mirnylab/cooltools) we calculated the tracks of contact insulation score. The method used is based on the algorithm described in (https://doi.org/10.1038/nature14450) and adjusted in (https://doi.org/10.1016/j.cell.2017.05.004). We calculated the insulation score as the total number of normalized and filtered contacts formed across that bin by pairs of bins located on the either side, up to 5 bins away, for each bin in our contact map binned at 4 kb resolution. Then we normalized the

score by its genome-wide median. To find insulating boundaries, we detected all local minima and maxima in the $\log_2$-transformed and then distinguish them by their prominence (Billauer E. peakdet: Peak detection using MATLAB, http://billauer.co.il/peakdet.html). The insulating boundaries were the detected minima in the insulation score, corresponding to a local depletion of contacts across the genomic bin. We empirically found that the distribution of log-prominence of boundaries has a bimodal shape. Based on that, we selected all boundaries in the high-prominence mode above a prominence cutoff of 0.1 for Hi-C mapped to the reference dm3 map, and a cutoff of 0.3 for allele-resolved Hi-C maps. We called the insulating boundaries in *thom* contact maps using the same approach and requiring a minimal prominence of 0.3. Finally, we removed boundaries that are adjacent to the genomic bins that were masked out during IC. We evaluated the similarity of insulating boundaries that were detected in the *cis* and *thom* contact maps by calculating the number of overlapping boundaries and allowing for a mismatch of up to four genomic 4 kb bins (16 kb total) between overlapping boundaries to account for the drift caused by the stochasticity of contact maps.

**Detecting tightly and loosely paired genomic bins**. We used the genome-wide PS track to classify each genomic bin as either tightly or loosely paired. We noticed that the genome-wide distribution of PS (Supplementary Fig. 6a, b) showed a well-pronounced peak at higher values of PS and a tail extending into the lower values of PS. We interpreted this distribution with a model, where each bin can be either tightly or loosely paired with a homologous locus on the second chromosome. In this model, tightly paired loci showed high PS values, producing the peak on the genome-wide distribution of PS, while loosely paired loci had lower PS values, giving rise to the tail of the PS distribution. We separated the peak from the tail of the distribution by fitting it with a sum of two Gaussians (Supplementary Fig. 6b); to stabilize the fitting procedure, we also clipped PS values below $-3$. The probability densities of the two Gaussians become the same at PS $= -0.71$. Thus, we classified all genomic bins with PS $< -0.71$ as loosely paired (since they are more likely to belong to the low-PS Gaussian) and the bins with PS $> -0.71$ as tightly paired.

**Determining tight and loose paired regions in the PnM genome**. A close examination of the PS track revealed two important features: (i) the fly genome is divided into regions that demonstrate consistently high, relatively similar, values of PS, followed by extended regions where PS dips into lower values, (ii) switching between high- and low-PS regions seemed to occur often around insulating boundaries.

We used the PS and insulating boundaries to determine the precision of pairing in the genome, and examine the variation of pairing more closely, in addition to the internal organization of tight and loose pairing. First, we divided the genome into regions between pairs of consecutive boundaries (as detected in reference-mapped, i.e., not allele-resolved Hi-C data). Second, we classified each of these regions as either tight or loosely paired, depending on the number loosely paired bins in that region. Because the dips in the PS track tend to be gradual, with PS being noticeably low only further away from the boundaries, we considered the cutoff of 25% of loosely paired bins per region to be sufficient to call the whole region as loosely paired. Finally, we noticed that this method occasionally split single loosely paired regions into a few smaller ones, presumably due to false positive boundary calls. To mitigate this issue, we merged consecutive loosely and tightly paired regions into larger ones if the boundary bin between them was classified the same way. We then used the detected regions of loose and tight pairing to calculate scaling curves $P_{cis}(s)^{loose}$, $P_{cis}(s)^{tight}$, $P_{thom}(s)^{loose}$, and $P_{thom}(s)^{tight}$. We calculated these curves using the same method as for the genome-wide $P_{cis}(s)$ and $P_{thom}(s)$, but only considering pairs of loci within the same region. See supplementary Note 1 for a detailed discussion.

**Quantifying genome-wide changes in pairing in knockdowns**. In order to quantify the genome-wide changes in pairing observed in knockdowns, we developed a metric that summarized the degree of pairing in each sample, and quantified the genome-wide changes in pairing observed in knockdowns, called Aggregated Pairing Score (APS):

APS = Mode (PS-CS), i.e., the most probable value of PS-CS genome-wide.

We reasoned that APS makes a good estimate for the degree of pairing exhibited by cells genome-wide—for two reasons. First, APS has a simple interpretation—since the PS and CS quantify the $\log_2$ intensity of the main diagonal, APS thus describes the most probable $\log_2$ ratio of short-distance *thom* contact frequency to short-distance *cis*. Second, APS is stable, since the most probable value in a distribution is not affected by the presence of heavy tails.

For a given pair of conditions, the statistical significance of the difference between their APS was determined by bootstrapping. Specifically, we tested the null hypothesis that the (PS-CS) distributions for both conditions were drawn from the same underlying distribution (i.e., if PS-CS for each condition has a similar distribution). We merged (PS-CS) distributions for both conditions, and then drew $1000\times$ pairs of random samples and calculated the difference of their APS. Finally, we calculated a *p*-value as a fraction of random sample pairs that had a larger difference of APS than the one observed in our data.

**Eigenvectors**. We quantified the compartment structure using of Hi-C maps with eigenvectors of observed/expected Hi-C maps, using a modified procedure from Imakaev et al.[74]. We performed eigenvector decomposition of the observed/expected 4 kb contacts maps subtracting 1.0 from each pixel. Finally, for each chromosome, we selected the eigenvector showing the highest correlation with the track of the number of genes overlapping each genomic bin. This method for eigenvector detection is available in *cooltools*.

**Loop quantitation**. The contact frequency at each loop was calculated as the sum of the Iteratively Corrected Contact Frequency in a 120 kb × 120 kb window surrounding the loop.

**ChIP-seq overlap with boundaries and Correlation with PS**. We mapped the publicly available raw ChIP-seq data following the procedure used by the ENCODE consortium[75], (https://github.com/ENCODE-DCC/chip-seq-pipeline). We used architectural protein ChIP datasets described in Supplementary Table 3.

After mapping ChIP-seq datasets to the reference genome, we called TF binding peaks using MACS2 with a maximal p-value cutoff of 0.01 and a minimal peak score cutoff of 200. Then, for each data set, we calculated (a) the percent of insulating boundaries in PnM cells overlapping ChIP-seq peaks using bedtools[76], and (b) the Spearman's rank correlation coefficient between the pairing score and the enrichment of the ChIP-seq signal over the input at 4 kb resolution.

**Determining *cis* and *thom* contact map resolution**. The estimated 4 kb resolution is based on ~75.4 million mappable pairs, with minor differences in the resolution for *cis* and *thom* contact maps. One of the key factors defining the resolution of a Hi-C dataset is the sequencing depth. If we pick a resolution too high (i.e., bin size is too small) for a given sequencing depth, we end up with many empty pixels (zero pixels), and overall, read counts would be distributed over many more pixels and thus would be more prone to sampling noise. In addition, because the mean number of read counts decays with distance, more distant diagonals will have more empty pixels. Thus, to address the difficulty of determining the optimal resolution across an entire contact map, we try different bin sizes and determine the number of empty pixels along each diagonal. The optimal resolution would be the one where there are only a few empty pixels at diagonals-of-interest—i.e., diagonals containing TADs (<~100 kb). In the case of untreated PnM cells, we looked at four resolutions: 1, 2, 4, and 10 kb and plotted the fraction of non-zero pixels in diagonals as a function of the genomic separation or distance (Supplementary Fig. 14). To interpret these curves, we set a criterion wherein the best resolution was the finest resolution at which we still had more than 50% non-zero pixels at 100 kb separation (relative average TAD scale). By this criterion, a 4 kb resolution emerged as optimal for both *cis* and *thom* contact maps.

**Detecting chr3R disomy**. During the analyses of the WGS and Hi-C reads, we noticed that the maternal homolog (057) of chromosome 3R had consistently higher sequencing coverage than the paternal one (439), from 3.24 Mb and to the telomere. This difference in sequencing coverage was particularly noticeable at the level of the IC balancing weights, which were on average 1.44X higher in this region for the paternal homolog (thus, compensating for lower coverage) comparing to the maternal homolog. Such pattern can be explained by a partial maternal disomy in a fraction of cells. A disomy in a fraction $x$ of cells leads to $r = (1 + x)/(1 - x)$ times higher sequencing coverage of the maternal homolog. Conversely, the observed ratio $r$ of sequencing coverage of the two homologs can be explained by partial uniparental disomy in $x = (r - 1)/(r + 1)$ fraction of cells. Using this approach, we estimate that the chr3R partial disomy is found in 17.9% cells of the untreated Hi-C sample, 20.6% cells of the mock-depleted sample, 21.5% of the Slmb-depleted sample and 19.6% of the TopII-depleted sample.

**RNA-Seq**. We mapped the raw RNA-seq data using STAR [https://academic.oup.com/bioinformatics/article/29/1/15/272537] following the same procedure as used by the ENCODE consortium [https://github.com/ENCODE-DCC/long-rna-seq-pipeline/tree/master/dnanexus].

**Reporting summary**. Further information on research design is available in the Nature Research Reporting Summary linked to this article.

## Data availability

The source data underlying Fig. 1d, Fig. 3e, and Supplementary Fig. 11a, and b are provided as a Source Data file. All raw sequencing data and extracted Hi-C contacts have been deposited in the Gene Expression Omnibus (GEO) repository under accession number GSE121256. Hi-C data obtained in this study is available for browsing using the HiGlass web browser[77]. Publicly available ChIP-seq datasets used in this study are listed in Supplementary Table 3.

## Code availability

All custom data analyses were performed in Jupyter Notebooks[78] using matplotlib [Hunter, John D. "Matplotlib: A 2D graphics environment." Computing in science &

engineering 9.3 (2007): 90–95.], numpy [Walt, Stéfan van der, S. Chris Colbert, and Gael Varoquaux. "The NumPy array: a structure for efficient numerical computation." *Computing in Science & Engineering* 13.2 (2011): 22–30.] and pandas [McKinney, Wes. "pandas: a foundational Python library for data analysis and statistics." *Python for High Performance and Scientific Computing* (2011): 1–9.] packages. We automated data analyses in command line interface using GNU Parallel [Tange, Ole. "Gnu parallel-the command-line power tool." The USENIX Magazine 36.1 (2011): 42–47.]. The software used in this study is available at https://github.com/mirnylab/.

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

## Acknowledgements

We thank the Wu and Mirny laboratories, participants of the Annual Northeast Regional Chromosome Pairing Conferences, the Lieberman Aiden laboratory, F. Bantignies, B. Beliveau, G. Filion, M. Francesconi, E. Joyce, B. Lehner, and J. Rowley for discussions, the TUCF Genomics Sequencing Core Facility for sequencing services, and the Drosophila Genomics Resource Center (NIH 2P40OD010949) for cell lines. We apologize to the authors whose work we could not cite due to constraints on referencing. This work was supported by awards from NIH/NIGMS (R01HD091797, R01GM123289, DP1GM106412), HMS to C.-t.W., EMBO (Long-Term Fellowship, ALTF 186-2014) to J.E., William Randolph Hearst Foundation to R.B.M., NIH Common Fund (R01 HG003143) to J.D. (Howard Hughes Medical Institute investigator), NIH/NIGMS (R01 GM114190) to L.A.M., J.D. and L.A.M. acknowledge support from the National Institutes of Health Common Fund 4D Nucleome Program (Grant U54 DK107980).

## Author contributions

J.A.A., J.E., and C.-t.W. designed the experiments. A.G. designed Hi-C computational analyses with input from J.A.A., J.E., L.A.M., and C.-t.W. J.D., and B.R.L. provided input in experimental design and data analysis. J.A.A., J.E., A.G., B.R.L., G.F., R.B.M., and W.S. carried out analyses of the Hi-C data. Experimental data were generated by J.A.A. and J.E. J.A.A. validated and characterized the cell lines, which were made by S.C.N. J.A.A. performed RNAi analyses. L.A.M. and G.F provided advice on Hi-C analyses. J.A.A., J.E., A.G., G.F, L.A.M., and C.-t.W. interpreted the data. J.A.A., J.E., A.G., L.A.M., and C.-t.W. wrote the paper with input from the other authors.

## Additional information

**Competing interests:** The authors declare no competing interests.

