## [Peer Review File · Nature Communications]

Reviewers' comments:

Reviewer #1 (Remarks to the Author):

The manuscript by Ting Wu and colleagues developed a haplotype-resolved Hi-C method by using a newly established hybrid cell line. This study provides the first whole-genome map of trans-homolog pairing in *Drosophila*. Authors showed that homolog pairing is the widespread mechanism that can occur as frequently as cis-contacts. Furthermore, authors suggested two types of homolog pairing; tight-pairing and loose-pairing. Active compartments tend to form tight pairing (Figure 3C and D), while loosely paired regions coincide with large topological domains (Figure 2H). Consistent with previous studies by the authors (Williams et al., *Genetics* 2007; Joyce et al., *PLoS Genet* 2012), RNAi knockdown of Slmb or Top II impairs the pairing efficiency in genome-wide. Overall, this study provides an important insight into our current understanding of genome organization. However, to warrant publication in *Nature Communications*, I would like to see additional mechanistic and functional analysis.

#1: Previous genetic studies suggested that insulator DNAs facilitate transvection (e.g., Fujioka et al., *PLoS Genet* 2016; Kravchenko et al., *MCB* 2005). Consistent with these observations, authors showed that boundary elements exhibit higher probability of homolog pairing than neighboring domains (Figure 4G). To explore the mechanism underlying homolog pairing in *Drosophila*, authors may want to address what types of insulator proteins (CTCF, Cohesin, Su(Hw), CP190, Mod(mdg4) etc.) are specifically enriched at the highly-paired boundaries.

#2: Authors suggest that transcriptionally active regions tend to form tight pairing (Figure 3C and D), while loosely paired regions coincide with large topological domains (Figure 2H). Based on this observation, they proposed a model in which pairing facilitates the formation of active transcriptional machineries across homologous chromosomes (Figure 4B). If a shared microenvironment is formed, one can imagine that maternal and paternal allele will show some coordination in their expression profiles. On the other hand, genes located in the loosely paired regions (or large topological domains) are expected to exhibit stochasticity. To warrant a model proposed by the authors, it would be important to address this point by further analyzing RNA-sequencing dataset.

#3: Authors used RNAi method to knockdown Slmb and Top II, which resulted in ~80% reduction of their mRNA level. However, if the turnover rate of these proteins were slow in this hybrid cells, the protein level might not be diminished at the comparable level. Indeed, Slmb RNAi in Kc167 cells has been reported to reduce pairing efficiency of Dodeca more than half (from ~70% to ~30%; Joyce et al., *PLoS Genet* 2012), while PnM cells exhibit relatively high frequency (~60%) even after the RNAi treatment (Figure 3E). To avoid ambiguity of the RNAi method, authors might want to test overexpression of CAP-H2, a target of the SCFSlmb complex. Previous study suggested that the simple overexpression of CAP-H2 can significantly change the pairing efficiency in *Drosophila* (Hartl et al., *Science* 2008; Smith et al., *Genetics* 2013).

Reviewer #2 (Remarks to the Author):

In this work, the authors first derived a *drosophila* cell line and then performed Hi-C experiment to study the homolog pairing. They also knocked down two genes by RNAi and studied their effect on pairing. Overall, I found the experiments are straightforward, but the data analysis is shallow.

Below are my major concerns:

1. The authors need to provide more stats on their Hi-C experiments: how many of the reads are long-range? How many reads can be used for phasing?
2. I am concerned about the quality of their experiments: in Supplementary Table. 2, only 12% of the reads are mappable (74million / 604 million); how good is the reference genome quality?
3. In Supp. Fig. 2, the percentages of tran-reads are dramatically different between the cell line and the embryo (26% vs 5%), what is the reason?
4. Did the authors call TADs and loops? For example, are there any paternal/maternal/trans-specific interactions?
5. In Fig. 2a, why trans-homolog signals is as strong as cis?
6. On the Hi-C map, what are the stripes between mat chr 2R to Mat/Pat chr 3R?
7. In Fig. 2C, trans-HiC, it seems there are still TAD-like structures. How to interpret it?
8. 36% loosely paired, 64% tightly paired ... how confident are the authors regarding these numbers? Can it be validated using other technologies?
9. Continue from point 8, this is population-based Hi-C, so the percentages are a mixer from different cells.
10. The authors didn't show chromatin interaction maps after the knock-down experiments.

Minor:

1. "We performed haplo-type resolved Hi-C .. this type of Hi-C .." The authors keep saying this in the title and many times in the main text: what type of Hi-C cannot be used for haplo-type phasing? I assume this is a standard in situ Hi-C, or something special?
2. The authors claimed "somatic pairing has now been implicated in numerous biological phenomena across a diversity of species, including mammals..." Can they specify what is known about pairing in mammal, say mouse or human?

Reviewer #3 (Remarks to the Author):

Summary:

In this manuscript, the authors first develop a diploid cell line appropriate for haplotype-resolved Hi-C and then use this cell line to understanding homolog pairing in *Drosophila*. Homolog pairing as a phenomenon has been described for a while, but the nature and structure of pairing is understudied and an unresolved problem in chromosome biology. The authors provide a new approach, haplotype-resolved Hi-C, to study this problem. With these points in mind, after minor revisions, this reviewer feels this manuscript is appropriate for publication in Nature Communications.

Major comments:

In the conclusions, page 15, lines 17-22: the authors seem to be discussing homolog pairing in the context of loop extrusion. However, earlier (page 11, lines 17-21) the authors point to pairing being more related to compartments than loop extrusion (see also minor comment below). Therefore, there is some inconsistency here that needs to be resolved as cohesin-dependent loop extrusion (Schwarzer et al., Rao et al.) is independent of compartmentalization. This should be resolved or clarified.

The author's claim that they reveal trans-homolog compartments, specifically that these show up as off-diagonal plaid patterns in the contact map (page 8, lines 1-3). It is hard to distinguish this plaid pattern in Extended Data Fig. 3a. The authors should show a Pearson's correlation map to clearly show the presence of A-B thom compartments.

Page 10, lines 8-13: Based on the $P_{cis}(s)$ curves, the authors are arguing that a transition in these curves suggests the presence of domains. What is unclear is how is it arrived at that for tightly paired regions this results in a series of small domains? How the authors are arriving at a series of, rather than the simple presence of, domains is not clear.

Minor comments:

Page 11, lines 17-21: The statement that pairing may be and is perhaps more related to compartments compared to loop extrusion is very astute. The authors may want to comment on the results of Rowley et al., *Mol Cell*, 2017 as well as high resolution *Drosophila* Hi-C datasets that do not observed CTCF-dependent chromatin looping. In addition to citation 38 on line 20, the authors should also cite Rao et al., *Cell*, 2017.

The author's Hi-C contact map is at 4 kb resolution. Is this based on all 75.4 M mappable pairs? If so, given that not all of these are thom reads, is there a resolution difference between the entire map and those specific to thom interactions?

Can the authors please clarify their claim on page 4, line 7 that "pairing can serve as a genome-wide regulatory mechanism"? What specifically is being regulated? Is this meant to be related to transvection and gene expression?

Reviewers' comments:

Reviewer #1 (Remarks to the Author):

The manuscript by Ting Wu and colleagues developed a haplotype-resolved Hi-C method by using a newly established hybrid cell line. This study provides the first whole-genome map of trans-homolog pairing in *Drosophila*. Authors showed that homolog pairing is the widespread mechanism that can occur as frequently as cis-contacts. Furthermore, authors suggested two types of homolog pairing; tight-pairing and loose-pairing. Active compartments tend to form tight pairing (Figure 3C and D), while loosely paired regions coincide with large topological domains (Figure 2H). Consistent with previous studies by the authors (Williams *et al.*, Genetics 2007; Joyce *et al.*, PLoS Genet 2012), RNAi knockdown of Slmb or Top II impairs the pairing efficiency in genome-wide. Overall, this study provides an important insight into our current understanding of genome organization. However, to warrant publication in Nature Communications, I would like to see additional mechanistic and functional analysis.

1. Previous genetic studies suggested that insulator DNAs facilitate transvection (e.g., Fujioka *et al.*, PLoS Genet 2016; Kravchenko *et al.*, MCB 2005). Consistent with these observations, authors showed that boundary elements exhibit higher probability of homolog pairing than neighboring domains (Figure 4G). To explore the mechanism underlying homolog pairing in *Drosophila*, authors may want to address what types of insulator proteins (CTCF, Cohesin, Su(Hw), CP190, Mod(mdg4) etc.) are specifically enriched at the highly-paired boundaries.

Response: Thank you for raising this very interesting point. We have now added citations for the involvement of insulator proteins in transvection and performed analyses using published ChIP-seq datasets to investigate the enrichment of insulator as well as other architectural proteins at PnM boundaries.

Excerpt (Introduction)

P4, line 5: "...In *Drosophila*, transvection has been observed at many loci, suggesting that pairing may even function as a regulatory mechanism genome-wide (Chen *et al.*, 2002, Kravchenko *et al.*, 2005, Bateman *et al.*, 2012a, Mellert and Truman, 2012, Fujioka *et al.*, 2016)."

Excerpt (Results):

P12, line 2: "...This distinction between tight and loose regions is also evident from visual inspection of the data (Fig 2f).

The association of higher pairing scores with domain boundaries (Fig. 2g) was particularly intriguing, given that domain boundaries are enriched in insulator and architectural proteins (Blanton *et al.*, 2003, Hou *et al.*, 2012, and reviewed by Chetverina *et al.*, 2014, Rowley and Corces, 2016), the observation that some insulator proteins and insulator elements promote transvection (Kravchenko *et al.*, 2005, Fujioka *et al.*, 2009, Fujioka *et al.*, 2016, bioRxiv, Piwko *et al.*, 2019 and reviewed by Geyer, 1997), and the enrichment of architectural proteins at genomic sites involved in allelic interactions (Rowley *et al.*, 2019). Indeed, despite a few discrepancies among different published ChIP-seq datasets, many insulator proteins were enriched at PnM

boundaries, with strong correlations between the ChIP-seq peaks and PS for some (e.g. Nup98, with the highest correlation coefficient) and a weak anti-correlation for others (e.g. Su(Hw), with the weakest correlation coefficient) (Supplementary Table 3). This analysis indicates a potential structural and functional regulatory role for some of these proteins on a genome-wide scale, such that, they would form insulated *thom* and *cis* domains and contribute to pairing.

Having elucidated the structure of paired homologs and its variation...”

Additional Response: In addition, when we did this analysis for architectural proteins and insulator proteins that were investigated in previous studies, we also found a strong correlation between Cap-H2 peaks and tight pairing (Spearman’s correlation coefficient = 0.449). We comment on this observation and discuss how it relates to tight pairing as well. See below for in-text additions.

Excerpt (Concluding remarks):

P19, line 11: “Finally, we turn to our observation that loci interacted with a second locus in *thom* nearly as often as they did in *cis*. Researchers have long speculated about the consequences of providing regulatory regions with a *cis-trans* choice (reviewed by *Apte et al, 2012 Joyce et al, 2016*), wherein pairing could enhance the co-regulation of allelic regions or enable transcriptional states to be transferred from one chromosome to another (*Lewis et al, 1954; Ashburner, 1967, Ashburner, 1977*). Indeed, as interactions in *cis* may preclude interactions in *trans*, the *cis-trans* choice recalls a hypothesis wherein regulation of a genomic region may require a balance, perhaps even a dynamic interplay, between pairing and unpairing, such as in a model counterbalancing pairing (linear locking) (*Wu et al, 1993*) with unpairing (looping or buckling out) (Fig. 4c). In other words, finely regulated genomic regions may need to be poised to pair or unpair on a moment’s notice. A capacity to transition easily between different states of pairing and unpairing may even promote or antagonize allele-specific expression, where the unpaired state may facilitate allele-specific expression, especially in mammals (reviewed by *Apte et al, 2012 and Joyce et al, 2016*). Homolog pairing may even accomplish what compartments do in both *cis* and *trans*, and what domains do in *cis*, co-localizing genomic regions to achieve an “economy of control (*Ashburner, 1977*). How structurally and functionally independent these processes are, remains to be explored.”

Excerpt (Figure 4 legend) P32, line 7: “...c, Left: Homologous loops between tightly paired regions may form by extrusion (black arrows), anti-pairing (blue arrows), or a combination of both. Right: Here, loops could result in *thom* domains that are either loosely paired (top), or railroad-track paired throughout if they fold back on each other, (bottom), with *cis*-maternal and *cis*-paternal domain boundaries concordant in both scenarios. Note, loop extrusion in mammalian systems is proposed to involve a cohesin ring (green oval) through which a single chromosome passes (*Sanborn et al., 2015, Fudenberg et al., 2016*), suggesting that extrusion in the context of pairing may involve the passage of each homolog separately (shown) or two homologs simultaneously (not shown), the outcome of which could be either loosely or tightly paired domains. In a nonexclusive alternative, loops are formed by anti-pairing (*Joyce et al, 2012*), where pairing might be counterbalanced with unpairing via anti-pairing factors such as Cap-H2. Interestingly, extrusion and/or anti-pairing could bring enhancers and promoters together at the base of the loops (indicated by *), activating transcription, such as might happen in tightly paired regions. In the context of anti-pairing, this

could explain the curious co-localization of Cap-H2 with regions of tight pairing (Supplementary Table 3; also (Rowley *et al.*, 2019))”

2. Authors suggest that transcriptionally active regions tend to form tight pairing (Figure 3C and D), while loosely paired regions coincide with large topological domains (Figure 2H). Based on this observation, they proposed a model in which pairing facilitates the formation of active transcriptional machineries across homologous chromosomes (Figure 4B). If a shared microenvironment is formed, one can imagine that maternal and paternal allele will show some coordination in their expression profiles. On the other hand, genes located in the loosely paired regions (or large topological domains) are expected to exhibit stochasticity. To warrant a model proposed by the authors, it would be important to address this point by further analyzing RNA-sequencing dataset.

Response: We very much share the Reviewer’s interest in the impact of pairing on the coordinated expression of alleles and are, indeed, planning to pursue this very issue. However, given that studies of allele-specific expression in our system would be best addressed with single-cell or clonal RNA-seq (Nag and Savova *et al.*, 2013), we are planning to explore this topic in depth in a subsequent, separate paper; as we believe that the ensemble RNA-seq in the current manuscript cannot easily answer questions regarding allele-specificity due to issues of cell-to-cell heterogeneity. That aside, the Reviewer’s comment has led us to realize our failure to properly address the potential issue of allelic coordination, and we thank him/her for pointing this out. We have thus made substantial changes to incorporate the important issue of allelic coordination within tightly paired regions. We also updated Figure 4b to reflect these changes.

Note that, during the consideration of the Reviewer’s comments, we also realized that we may have inadvertently left out the discussion on the consequences of loosely paired regions. Hence, we have added a section that balances our discussion of tightly and loosely paired regions.

Excerpt (Results) P13, line 8: “...Excitingly, these trends were confirmed globally, with active regions enriched for high PS, heterochromatin (green) showing a bimodal distribution, and repressed (blue) and inactive (black) chromatin containing regions of both high and low PS (Fig. 3b).”

Excerpts (Concluding remarks) P17, line 14: “...We also examined the relationship between pairing and genome function, discovering that tight pairing can be correlated with either expressed or repressed regions, while loose pairing is correlated primarily with repression or inactive chromatin. While this finding may suggest that gene activity can, but does not always, promote tight pairing, it is also possible that tight pairing facilitates the formation of microenvironments that can, but do not always, favor transcription. Such microenvironments may promote the entangling of R-loops (Fig. 4bi) or enrichment of RNA polymerase, transcription factors (Lim *et al.*, 2018, Jackson *et al.*, 1993, Edelman and Fraser, 2012, Hilbert *et al.*, 2018, and/or insulator elements and associated proteins at domain boundaries (Dixon *et al.*, 2012, Tang *et al.*, 2015, Hou *et al.*, 2012, Ulianov *et al.*, 2016, Lim *et al.*, 2018, Rowley *et al.*, 2019). In brief, a pairing-mediated microenvironment may result in a more robust level of either expression or repression (Fig. 4bi, ii) (Supplementary Table 4), consistent with the association of transvection with both gene activation and gene repression (reviewed by (McKee, 2004, Apte and

Meller, 2012, Kassis, 2012, Fukaya and Levine, 2017, Joyce et al., 2016). It is also possible that the different types of pairing promote or antagonize allele-specific expression; further investigation of these scenarios will require single-cell analyses. Note that our findings differ from predictions of a study..."

P18, line 10: “Loosely paired regions are equally interesting. First, unlike tightly paired domains, which are associated with both active as well as repressed regions, loosely paired domains show a preference for repressed genomic regions, with just a small percentage of the genome being both loosely paired and expressed (Fig. 3c). While these observations may suggest that inactive genomic regions lead to loose pairing, it is also possible that loose pairing is inherently not permissive of transcription. If the latter were true, and speculating broadly, then it may be that achieving tight pairing could be a first step in becoming susceptible to regulation at some loci.

P18, line 18: “Loosely paired regions are also interesting because they lack a *thom* diagonal, indicating lack of railroad track pairing within some *thom* domains (Fig. 2f, schematics below). Importantly, the boundaries of these loosely paired regions are tightly paired, supporting a model that integrates pairing, loop formation, and chromosome compaction via a mechanism wherein chromosomes are looped (buckled out) by anti-pairing between regions of pairing (Joyce et al., 2012). In these loosely paired regions, homologs (and, perhaps, sister chromatids) could be extruded or formed via some other mechanism (Fudenberg et al., 2018) and/or anti-paired between tightly paired regions but still interact by virtue of remaining tightly paired at their loop bases. In this scenario, tightly paired regions could behave as extrusion barriers and become boundaries (Fig. 4c). Interestingly, RNA polymerase and insulator proteins have been proposed to behave as barriers to extrusion and thus may be in play in this scenario (Supplementary Table 3) (*bioRxiv* (Brandão et al., 2019) and reviewed by (Hug and Vaquerizas, 2018)). Lack of a diagonal has also been observed for polytenized chromosomes (Eagen et al., 2015, Kolesnikova, 2018), where it may reflect an outnumbering of *cis* contacts by an abundance of *trans* contacts. These observations raise the possibility that, under some circumstances, there may be competitive relationships between short-range *cis* and *thom* contacts (Wu, 1993, Joyce et al., 2016) and/or between short-range and long-range *thom* contacts.”

Excerpt (Figure legends) P32, line 2: “Figure 4. Haplotype-resolved Hi-C in PnM cells distinguished different forms of *thom* from *cis* interactions a, *Cis* contact maps for two homologs and schematics depicting possible *cis* interactions. b, *Thom* contact map demonstrating a variable structure of pairing, including tight, precise pairing interspersed with looser, less precise pairing. *Thom* interactions encompass organizational structures that are concordant with *cis* behavior and could facilitate a variety of transcriptional states, including i. active or ii. repressive environments...”

3. Authors used RNAi method to knockdown Slmb and Top II, which resulted in ~80% reduction of their mRNA level. However, if the turnover rate of these proteins were slow in this hybrid cells, the protein level might not be diminished at the comparable level. Indeed, Slmb RNAi in Kc167 cells has been reported to reduce pairing efficiency of Dodeca more than half (from ~70% to ~30%; Joyce et al., PLoS Genet 2012), while PnM cells exhibit relatively high frequency (~60%) even after the RNAi treatment (Figure 3E). To avoid ambiguity of the RNAi

method, authors might want to test overexpression of CAP-H2, a target of the SCFSlmb complex. Previous study suggested that the simple overexpression of CAP-H2 can significantly change the pairing efficiency in *Drosophila* (Hartl *et al.*, Science 2008; Smith *et al.*, Genetics 2013).

Response: We agree with the Reviewer that other studies have shown a stronger disruption of pairing, such as those conducted in Kc₁₆₇ cells. The differences in pairing disruption between PnM and other cell types could be due to a more rapid protein turnover or a lower transfection efficiency in PnM. These differences may also be due to a slightly different nature of pairing in tetraploid in Kc₁₆₇ or S2 culture cells; note that Kc₁₆₇ and S2 cells are variably tetraploid, which not only may affect pairing but likely also complicates the interpretation of FISH assays. In fact, in addition to PnM, other diploid culture cells, such as clone 8, have also shown a lower disruption of pairing as a result of RNAi (Senaratne *et al.*, 2017). It may also be worth noting that the greatest disruption of pairing in clone 8 cells was achieved using chemical inhibition of TopII (Williams and Bateman *et al.*, 2007). With that in mind, we also chemically inhibited TopII in PnM cells using ICRF-193 and found that it did not lead to significant disruption of pairing (data not shown).

Thus, to address the Reviewer's comments, we overexpressed a venus-tagged Cap-H2 under an Actin promoter (via Avw-Cap-H2-WT) to disrupt pairing and address the possibility of slow Slmb turnover in Slmb RNAi experiments. Results showed that, despite overexpression of Cap-H2 (Extended Data Fig. 11b), levels of pairing were not affected significantly. This outcome could be due to lower levels of venus-Cap-H2-WT protein, as compared to those expressing Cap-H2 in previous studies (Hartl *et al.*, 2008, Smith *et al.*, 2013).

Excerpt (Results) P15, line 18: “Excitingly, using RNAi to target two genes known to promote pairing, Slmb (component of SCF^{Slmb} complex (Buster *et al.*, 2013, Hartl *et al.*, 2008, Joyce *et al.*, 2012)) and Topoisomerase II ((TopII (Williams *et al.*, 2007))), we reduced the corresponding mRNA levels in PnM cells by $75.2 \pm 2.8\%$ and $82.5 \pm 5.0\%$, respectively (Extended Data Fig. 10a). Importantly, we observed a concomitant 10.7-12.1% reduction of pairing as assayed by FISH (Fig. 3e; Supplementary Table 5). While this reduction was less than previously reported (Joyce *et al.*, 2012, Williams *et al.*, 2007), it was significant as compared to mock RNAi trials ($P < 0.05$) (Fig. 3e). We also attempted a stronger disruption of pairing by overexpressing Cap-H2 (Hartl *et al.*, 2008, Smith *et al.*, 2013), but found that levels of pairing were not affected significantly at those loci (Extended Data Fig. 11a). Since knockdown experiments were more disruptive of pairing, we generated Hi-C maps for the knockdown and mock experiments, each with about 20 million haplotyped mappable reads (Supplementary Table 2) and found a reduction in $P_{thom}(s)/P_{cis}(s)$ for both Slmb and TopII RNAi samples at all separations as compared to mock and untreated sample (Extended Data Fig. 10b; error bars for each sample fall within lines). Note that, the values of $P_{thom}(s)/P_{cis}(s)$ for the mock samples veer below those for untreated controls at genomic separations greater than 100 kb. While this reduction suggests that the knockdown treatment may perturb *thom* and/or *cis* interactions and thus may be interesting in and of itself, our focus has been on the even greater reduction in $P_{thom}(s)/P_{cis}(s)$ for both RNAi-treated samples (Extended Data Fig. 10b). Slmb and TopII knockdowns also produced a change in PS. To quantify this change, we computed the aggregated pairing score (APS) as the mode of (PS-CS) distribution, which summarizes the degree of pairing with a single value (Supplementary methods). As shown in Figure 3f, APS dropped after

knockdown of Slmb or TopII, as compared to mock, and the untreated sample. These observations were consistent across replicates (Extended Data Fig. 10c) and across tight and loose regions (Extended Data Fig. 12; Supplementary methods). In addition, some *thom* interaction peaks in Slmb and TopII RNAi samples were depleted as compared to Mock sample (Extended Data Fig. 13). In summary, not only were PnM cells amenable to RNAi, but Hi-C could detect global changes in pairing as a result of the knockdown of pairing factors.”

Reviewer #2 (Remarks to the Author):

In this work, the authors first derived a drosophila cell line and then performed Hi-C experiment to study the homolog pairing. They also knocked down two genes by RNAi and studied their effect on pairing. Overall, I found the experiments are straightforward, but the data analysis is shallow.

Below are my major concerns:

1. The authors need to provide more stats on their Hi-C experiments: how many of the reads are long-range? How many reads can be used for phasing?

Response: We agree with the Reviewer that providing additional read pair counts for our experiments will be informative and so added two tables to the supplement:

Supplementary Information file, P55: Supplementary Table 10 shows the total number of raw reads representing the PnM, maternal (DGRP-057), and paternal (DGRP-439) genomes, in addition to reads that were used for phasing and calling SNVs *de novo* (PnM: 131,879,436, DGRP-057: 58,974,711, DGRP-439: 58,287,559).

Supplementary Information file, P56: Supplementary Table 11 shows the breakdown of haplotype-resolved read pairs for each replicate at different genomic separations. While the number of *thom* read pairs at 3 kb genomic separation are 5,548,717 and 5,769,358 for replicates 1 and 2, respectively, those at 100 kb genomic separation are 2,516,559 and 2,652,904.

2. I am concerned about the quality of their experiments: in Supplementary Table. 2, only 12% of the reads are mappable (74million / 604 million); how good is the reference genome quality?

Response: The Reviewer’s concern brings up an important detail regarding the statistics of the mappability of our read pairs. In particular, the ~12% mappability reflects the mapping of haplotype-specific read pairs. Basically, in order for a read pair to be successfully mapped, it must overlap at least one SNV on each of its two sides. This requirement produces a significant reduction in the efficiency of mapping. In contrast, the mappability of non-haplotype resolved read pairs to the reference genome is much higher (~68-70%). We added these statistics to Supplementary Table 2 and provided a better explanation of haplotype mapping in the text:

Excerpt (Results) P6, line 18: “We next performed haplotype-resolved Hi-C on PnM cells, an approach which separates, *in silico*, the read pairs into five categories: *cis*-maternal, *cis*-paternal, *trans*-homolog, *trans*-heterolog, and unresolvable by haplotypes. In this form of Hi-C,

each of the two fragments of genomic DNA that are ligated together by virtue of their proximity *in situ* are assigned a parental origin based on the SNVs they carry, thus permitting researchers to distinguish Hi-C read pairs that represent *cis*-maternal, *cis*-paternal, *trans*-homolog (*thom*), and *trans*-heterolog (*thet*) interactions. By selecting only those read pairs with at least one SNV per side, we obtained 75.4 million mappable read pairs, producing a 4 kb resolution haplotype-resolved map of the mappable portion of the genome (e.g., excluding repetitive regions), wherein less than 0.4% of *thom* read pairs are expected to have resulted from read misassignment (Supplementary methods; Extended Data Fig. 2a, b, Supplementary Table 2). This gave us great confidence in our ability to select haplotype-specific reads, and then map them to the hybrid PnM genome.”

3. In Supp. Fig. 2, the percentages of trans-reads are dramatically different between the cell line and the embryo (26% vs 5%), what is the reason?

Response: We thank the Reviewer for pointing out a need clarify why our cell line supports a level of pairing that is higher than that observed in early embryos. In brief, pairing does not reach maximum levels in early *Drosophila* embryos because early embryogenesis is the developmental time period when homolog pairing is just initiating. Accordingly, we edited the text as follows:

Excerpt (Results) P7, line 11: “... Strikingly, however, *thom* read pairs were ~7.8 times more abundant in PnM cells than in *Drosophila* embryos (Extended Data Fig. 2b). In addition, when considering *thom* contacts as a function of the separation of loci along the genome (genomic separation), we found them to be more abundant at all genomic separations (Extended Data Fig. 2c, d). These observations are not surprising as they agree with the higher levels of pairing observed by FISH in PnM cells (Fig. 1d) as compared to developing embryos where pairing is just initiating (*Fung et al., 1998*), *Erceg, AlHaj Abed, Goloborodko et al, 2018, bioRxiv*), possibly due to a greater percentage of cells with paired homologs, an increased fraction of the genome exhibiting pairing, and/or a smaller proportion of dividing cells in the PnM cell line (Extended Data Fig. 3). Importantly, the greater abundance of *thom* contacts argued that an analysis of pairing in PnM cells would yield new insights into the structure of paired homologs.”

4. Did the authors call TADs and loops? For example, are there any paternal/maternal/trans-specific interactions?

Response: We did call loops and *cis* and *thom* TADs (referred to as domains throughout the text). For domain calling, we calculated the contact insulation score, and then used these scores to find insulating boundaries of domains (described in the Supplementary methods in detail). As for loop calling, the standard loop calling algorithms are designed for mammalian data and, thus, may not work well in flies, especially since the latter has cohesin-independent loops (*Eagen et al, 2017*, and *Ogiyama et al, 2018*). However, we annotated loops manually as in *Stadler et al, 2017*, and *Eagen et al, 2017*, despite having deeply sequenced Hi-C *Drosophila* data. Regarding allele specific interactions, the Reviewer brings up a very interesting point which we address below.

First, with respect to domains, as we showed in Figure 2d, 81.5% and 89.1% of the domain boundaries in the *cis* and *thom* maps appeared in the *thom* and *cis* maps, respectively. In our companion paper, we also noted that there is a high degree of overlap in maternal and paternal *cis* insulating boundaries (Erceg, AlHaj Abed, Goloborodko et al, 2018, bioRxiv; *Extended data Fig. 5d*). Second, with respect to loops, in our PnM cell line data, we observe ~70 loops or interaction peaks in *cis* contact maps and, although the majority of those loops are detected in maternal, paternal, and *thom* contact maps, some are not present across all three contact maps. These differences may have functional implications but will require thorough analysis to validate maternal- or paternal- specific interactions and, we feel, lies beyond the goals of this study. However, to address the Reviewer’s concerns, we have added a few examples of loops or interaction peaks that appeared in all three maps (*Extended data Fig 5a*) as well as a few that were detected in either *cis*-maternal, *cis*-paternal, or *thom* contact maps (*Extended data Fig 5b*) and then adjusted the text accordingly:

Excerpt (Results) P9, line 5: “Besides the general distance-dependent decay of *cis* and *thom* contact frequency, our Hi-C maps also revealed a rich structure of *thom* interactions, including well-defined *thom* domains, at genomic separations as small as tens of kilobases, loops or interaction peaks, as well as plaid patterns of contacts far off the diagonal, at genomic separations as large as tens of megabases and corresponding to compartments (*Extended Data Fig. 4a, Extended Data Fig. 5a*). Consistent with railroad track pairing, we found strong concordance between the *thom*, *cis*-maternal, and *cis*-paternal Hi-C maps in terms of the positions and sizes of domains and loops (*Fig. 2c*, with Hi-C diagonal positioned horizontally; *Extended Data Fig. 4b-d, Extended Data Fig. 5a*), although with some exceptions (*Extended Data Fig. 5b*). In addition, 81.5% and 89.1% of the domain boundaries in the *cis* and *thom* maps appeared in the *thom* and *cis* maps, respectively (*Fig. 2d*). Overall, the strong concordance between *thom*, *cis*-maternal, and *cis*-paternal Hi-C maps indicated a high level of registration between paired homologs.”

5. In *Fig. 2a*, why trans-homolog signals is as strong as *cis*?

Response: We share the Reviewer’s surprise and excitement with respect to the observation in *Fig 2a*. As we look at the genome-wide map, the *thom* interactions are so abundant that they are almost as frequent as are *cis* interactions, reflecting what we described in the text as “railroad type pairing”. In more technical terms, this observation is best described by the curve of the contact frequency $P(s)$ relative to genomic separation (*Fig 2b*), in which *thom* contact frequency is almost equivalent to *cis* at most genomic separations. For example, at genomic separations of 10 and 100 kb, the ratio of *thom*-to-*cis* interactions is 0.7 and 0.9, respectively. The similarity of *thom* and *cis* interaction frequency reflects the high degree of pairing between homologous chromosomes in our PnM cell line.

6. On the Hi-C map, what are the stripes between mat chr 2R to Mat/Pat chr 3R?

Response: We thank the Reviewer for pointing out the need to explain this observation. These *trans*-heterolog interactions appear due to the clustering of sub-telomeric regions of different chromosome arms. We added a clarification of this observation in the text.

Excerpt (Results) P7, line 8: “As shown in Figure 2a, homologs are aligned genome-wide, comparable to the global *thom* signature detected in early *Drosophila* embryos (Erceg, AlHaj Abed, Goloborodko et al, 2018, *bioRxiv*. In addition, *trans*-heterolog interactions are detected globally and include sub-telomere clustering (e.g 2R to 3R) (Mizuguchi et al., 2014, Hug et al., 2017, Stadler et al., 2017)...”

7. In Fig. 2C, trans-HiC, it seems there are still TAD-like structures. How to interpret it?

Response: Again, we share the Reviewer’s surprise with regard to this observation. In fact, we believe this is one of the most exciting observations in our study, where some *thom* domains preserve the structure, shape and diagonal observed in domains in *cis*, while others don’t. Our interpretation is that pairing (in the domains that look identical to *cis* domains) is tight, precise and in-register. We added a statement in the text to emphasize the contrast between the different domain types observed in *thom*:

Excerpt (Results) P10 line 11: “...Lack of a diagonal may reflect any number of structures, including imprecise and/or loose pairing or even the side-by-side alignment of homologous, yet distinguishable, domains. In contrast, domains retaining the structure and diagonal observed in corresponding *cis* domains may represent railroad pairing throughout the domains (Fig. 2f, schematics below).”

8. 36% loosely paired, 64% tightly paired ... how confident are the authors regarding these numbers? Can it be validated using other technologies?

9. Continue from point 8, this is population-based Hi-C, so the percentages are a mixer from different cells.

Response: The Reviewer raises interesting points with regard to the reproducibility of our approach for estimating tight and loose pairing as well as the issue of population-based Hi-C. One way to validate our results would be to determine the level of pairing at a large number (~10-20 regions across the genome) of tightly and loosely paired regions using FISH-based imaging, including perhaps measurements of the physical distance between homologs using super-resolution imaging. Wide-field imaging has a resolution limit of ~200 nm, which may be a limitation to distinguishing tight/loose pairing. In fact, we are working toward exactly these experiments. However, as they are quite complicated, we are hoping to include the anticipated findings in a subsequent paper.

Regarding the reproducibility of our bioinformatics approach in determining the percent of tight-to-loose ratio, we are confident of the existence of tight and loose regions in our Hi-C maps for PnM cells and report that the ratio of tightly to loosely paired regions was determined

by fitting two Gaussians, the tight-to-loose ratio varies depending on exactly where the cutoff is drawn. We determined this cutoff automatically and, in repeating the analysis on a bin-by-bin basis on each of our untreated PnM replicates, we get a threshold cut-off of -0.74 for replicate 1 and -0.77 for replicate 2 (Extended Data Fig. 6b). These thresholds resulted in similar tight-to-loose ratios for replicates 1 and 2, respectively (Extended Data Fig. 6d).

Excerpt (Results) P10, line 17: “To better understand genome-wide variation in pairing, we examined the PS distribution (Extended Data Fig. 6a) and noted that it could be approximated by two normal distributions that were reproducible for each replicate (Extended Data Fig. 6b, d). These distributions suggested two classes of loci, one consisting of more tightly paired (higher PS) loci and the other consisting of more loosely paired (lower PS) loci, defined using only a single cut-off (PS = -0.71) (Extended Data Fig. 6b). While such a deconvolution likely oversimplifies the reality of pairing we nevertheless used it to bootstrap our investigation forward. Specifically, we divided the Hi-C amenable portion of the whole genome into regions of tight and loose pairing by first classifying each domain as either tightly or loosely paired based on its PS, and then merging consecutive domains of the same pairing type into one region (Extended Data Fig. 7a, b; see Supplementary methods). According to this classification procedure, ~34% of the genome is loosely paired, and ~66% is tightly paired (Extended Data Fig. 6c, d).”

10. The authors didn't show chromatin interaction maps after the knock-down experiments.

Response: We thank the Reviewer for encouraging us to include these data. Indeed, the corresponding contact maps were able to reveal changes after knockdown and, thus, we added a figure showing KD maps for two chromosome arms in Extended Data Fig. 13. These interaction maps reveal changes in a few long-range interactions, or interaction peaks, which we have quantified in each of the treatments (Slmb, and TopII) and the control in Extended Data Fig. 13. We refer to that figure as shown in the text below:

Excerpt (Results) P16, line 11: “Slmb and TopII knockdowns also produced a change in PS. To quantify this change, we computed the aggregated pairing score (APS) as the mode of (PS-CS) distribution, which summarizes the degree of pairing with a single value (Supplementary methods). As shown in Figure 3f, APS dropped after knockdown of Slmb or TopII, as compared to mock, and the untreated sample. These observations were consistent across replicates (Extended Data Fig. 10c) and across tight and loose regions (Extended Data Fig. 12; Supplementary methods). In addition, some *thom* interaction peaks in Slmb and TopII RNAi samples were depleted as compared to Mock sample (Extended Data Fig. 13). In summary, not only were PnM cells amenable to RNAi, but Hi-C could detect global changes in pairing as a result of the knockdown of pairing factors.”

Minor:

1. “We performed haplo-type resolved Hi-C .. this type of Hi-C ..” The authors keep saying this in the title and many times in the main text: what type of Hi-C cannot be used for haplo-type phasing? I assume this is a standard in situ Hi-C, or something special?

Response: We thank the Reviewer for making us aware of where the explanation of our approach fell short. What we should have pointed out is that haplotype-resolved Hi-C maps differ from those that are not resolved into haplotypes simply because the former separates, *in silico*, the read pairs into five categories: *cis*-maternal, *cis*-paternal, *trans*-homolog, *trans*-heterolog, and unresolvable by haplotypes. Thus, the difference lies in the whether the analysis of the read pairs includes consideration of SNVs. From an experimental perspective, the Hi-C we used in this paper is the same as the one developed by Rao *et al*, 2014, and is indistinguishable from the standard *in situ* Hi-C, which can be used for phasing in mammals and especially since *cis* interactions are much more abundant than any *trans* interactions there. In *Drosophila*, since *trans*- interactions are much more abundant, we resort to haplotype-specific Hi-C, or allele-specific Hi-C for deducing *cis* and *thom*-specific interactions. We have now made this clear through the following change(s) to the text:

Excerpt (Results) P6, line 18: “We next performed haplotype-resolved Hi-C on PnM cells, an approach which separates, *in silico*, the read pairs into five categories: *cis*-maternal, *cis*-paternal, *trans*-homolog, *trans*-heterolog, and unresolvable by haplotypes. In this form of Hi-C, each of the two fragments of genomic DNA that are ligated together by virtue of their proximity *in situ* are assigned a parental origin based on the SNVs they carry, thus permitting researchers to distinguish Hi-C read pairs that represent *cis*-maternal, *cis*-paternal, *trans*-homolog (*thom*), and *trans*-heterolog (*thet*) interactions...”

2. The authors claimed “somatic pairing has now been implicated in numerous biological phenomena across a diversity of species, including mammals...” Can they specify what is known about pairing in mammal, say mouse or human?

Response: We added a more explicit statement regarding pairing in mammals:

Excerpt (Introduction) P3, line 23: “...Somatic pairing has now been associated with numerous biological phenomena across many species, including mammals, where pairing has been implicated in DNA repair, X-inactivation, imprinting, V(D)J recombination, and the establishment of cell fate (reviewed by (Joyce et al., 2016)). Pairing-dependent gene regulation, a well-recognized form of transvection, is among the best understood of biological phenomena associated with pairing (reviewed by (McKee, 2004, Apte and Meller, 2012, Kassiss, 2012, Joyce et al., 2016, Fukaya and Levine, 2017))...”

Reviewer #3 (Remarks to the Author):

Summary:

In this manuscript, the authors first develop a diploid cell line appropriate for haplotype-resolved Hi-C and then use this cell line to understanding homolog pairing in *Drosophila*. Homolog pairing as a phenomenon has been described for a while, but the nature and structure of pairing is understudied and an unresolved problem in chromosome biology. The authors provide a new approach, haplotype-resolved Hi-C, to study this problem. With these points in mind, after minor

revisions, this reviewer feels this manuscript is appropriate for publication in Nature Communications.

Response: We thank the Reviewer for this encouraging comment!

Major comments:

1. In the conclusions, page 15, lines 17-22: the authors seem to be discussing homolog pairing in the context of loop extrusion. However, earlier (page 11, lines 17-21) the authors point to pairing being more related to compartments than loop extrusion (see also minor comment below). Therefore, there is some inconsistency here that needs to be resolved as cohesin-dependent loop extrusion (Schwarzer *et al.*, Rao *et al.*) is independent of compartmentalization. This should be resolved or clarified.

Minor comments:

2. Page 11, lines 17-21: The statement that pairing may be and is perhaps more related to compartments compared to loop extrusion is very astute. The authors may want to comment on the results of Rowley *et al.*, Mol Cell, 2017 as well as high resolution Drosophila Hi-C datasets that do not observed CTCF-dependent chromatin looping.

In addition to citation 38 on line 20, the authors should also cite Rao *et al.*, Cell, 2017.

Response: We thank the Reviewer for this encouragement to expound a bit further on our thinking with regards to the topics of compartments, domains, loop extrusion, and their relationship to pairing. Although there is no simple interpretation of the relationship between pairing and these structures, we have tried to address the discrepancies and address the work by Rowley *et al.*, 2017.

In addition, Schwarzer *et al.*, 2017 refers to a study on the independence of loop formation from compartment formation and not on cohesin-dependent loop extrusion. For that reason, the citation did not include Rao *et al.*, 2017, Sanborn *et al.*, 2015, or Fudenberg *et al.*, 2016. We reworded that sentence to avoid confusion below:

Excerpt (Results) P13, line 12: “Next, we examined the relationship between pairing and the 3D spatial compartmentalization of active and inactive chromatin (Lieberman-Aiden *et al.*, 2009). Here, we observed a strong correlation between high PS values and the *cis* eigenvector track (a measure of compartments as determined from Hi-C maps) in individual genomic regions (Extended Data Fig. 8) as well as genome-wide ($r_s = 0.71$, $p < 10^{-10}$; Fig. 3c). Regions with high PS values and thus likely to be tightly paired were in predominantly active A-type compartments (54.4% of mappable genome) as versus inactive B-type compartments (12.6%). Conversely, regions with lower PS values and thus likely to be loosely paired were more often in B-type (25.2%) as versus A-type (7.9%) compartments (Fig. 3C). In short, homolog pairing was correlated with compartmentalization of the genome, and active A-type compartments were more likely to be tightly paired. As compartmentalization of the genome into active and inactive compartments may be independent of TAD formation in mammals (Schwarzer *et al.*, 2017), our

observations may suggest that, pairing may be more related to compartments, gene expression, and the epigenetic states governing them. Taken together with the recent report on the major role of compartmentalization in *Drosophila cis* genome architecture (Rowley et al., 2017), these observations put compartmentalization as the main force behind formation of both *cis* and *thom* genome architecture.”

Additional response: With regards to the statements in the conclusion, we believe different structures (e.g., compartments, domains, or loops) facilitate efficient functional utilization of the genome. This can be achieved either through the formation of domains (e.g., insulation by boundaries from other domains) or through the segregation of the genome into different compartments based on transcriptional activity and epigenetic state. Our concluding statement emphasizes how the structural design of pairing could facilitate efficient function.

Excerpt (Concluding remarks) P19, line 22: “...Homolog pairing may even accomplish what compartments do in both *cis* and *trans*, and what domains do in *cis* (Wu, 1993), co-localizing genomic regions to achieve an “economy of control (Ashburner, 1977)”. How structurally and functionally independent these processes are, remains to be explored.”

3. The author’s claim that they reveal trans-homolog compartments, specifically that these show up as off-diagonal plaid patterns in the contact map (page 8, lines 1-3). It is hard to distinguish this plaid pattern in Extended Data Fig. 3a. The authors should show a Pearson’s correlation map to clearly show the presence of A-B thom compartments.

Response: We agree with the Reviewer and have now provided a Pearson’s correlation for A-B compartmentalization for the *thom* contact map in Extended Data Fig 4a, P33 of Supplementary Information.

4. Page 10, lines 8-13: Based on the $P_{cis}(s)$ curves, the authors are arguing that a transition in these curves suggests the presence of domains. What is unclear is how is it arrived at that for tightly paired regions this results in a series of small domains? How the authors are arriving at a series of, rather than the simple presence of, domains is not clear.

Response: The Reviewer highlights an important issue. To better understand the structures of tightly and loosely paired regions, we selected large regions with a relatively narrow size range of 200-400 kb or 100-200 kb. As a result, the structures we observe are within that size range. We address it in the text as follows:

Excerpt (Results) P11, line 7: “To better understand chromosome organization within tightly and loosely paired regions, we selected those spanning distances large enough for us to conduct our studies (200-400 kb or 100-200 kb; Supplementary methods, Extended Data Fig. 7) and calculated $P_{thom}(s)$ and $P_{cis}(s)$. Tightly and loosely paired regions differed in the decay of *cis* and *thom* contact frequencies. Within the 200-400 kb tightly paired regions, *thom* contacts at the highest registration (smallest genomic separation, $s = 1$ kb) appeared as frequent as *cis* contacts at $s = \sim 5$ kb. In loose regions, the frequency of such *thom* contacts matched that of *cis* contacts at $s = \sim 30$ kb (Fig. 2h, marked on graph). This indicated that, in loose regions, homologs were aligned less precisely. Surprisingly, we found that regions of tight and loose pairing also differed in their internal organization. This was evident from the different shapes of their $P_{cis}(s)$ curves –

in tight regions, the $P_{cis}(s)$ curve had two modes (Fig. 2h, left), a shallow mode at $s < \sim 30$ kb and a steep mode at $s > \sim 30$ kb, while in loosely paired regions, we observed only a shallow mode (Fig. 2h, right). Drawing from other Hi-C studies, where the presence of a shallow mode followed by steep mode is a signature of domains (Schwarzer et al., 2017, Fudenberg et al., 2018), we then further interpreted our *cis* data. In particular, the transition of $P_{cis}(s)$ at ~ 10 -30 kb within 200-400 kb tightly paired regions, suggested that they consisted of a series of relatively small domains, within which pairing may reflect primarily the constraints imposed by tight pairing at the boundaries. In contrast, we did not see a similar transition of $P_{cis}(s)$ within these 200-400 kb loosely paired regions, suggesting that each of these regions constituted a single domain. This distinction between tight and loose regions is also evident from visual inspection of the data (Fig. 2f).”

5. The author’s Hi-C contact map is at 4 kb resolution. Is this based on all 75.4 M mappable pairs? If so, given that not all of these are *thom* reads, is there a resolution difference between the entire map and those specific to *thom* interactions?

Response: We appreciate the Reviewer’s questions regarding how we arrived at a 4 kb resolution for our haplotype-resolved contact maps. Accordingly, we now make clear that the estimated 4 kb resolution is indeed based on ~ 75.4 million mappable pairs (with minor differences in the resolution for *cis* and *thom* contact maps) as well as explain how we settled on this resolution. In brief, we explain how, if we had picked a resolution that was too high (i.e., bin size is too small) for a given sequencing depth, read counts would be distributed over many more pixels, and we would have been more prone to sampling noise. Thus, to determine the optimal resolution across the entire contact map (including *cis* and *thom* contacts), we tried different bin sizes (i.e., different resolutions), determined the number of empty pixels along each diagonal, and selected the bin size giving the best coverage (few empty pixels). By this criterion, a 4 kb resolution was optimal for both for *cis* and *thom* contact maps. We have now added Extended Data Fig 14 and a section in our supplementary methods expanding on our rationale, as shown below:

Excerpt (Supplementary methods) P24, line 12: Determining *cis* and *thom* contact map resolution

“The estimated 4 kb resolution is based on ~ 75.4 million mappable pairs, with minor differences in the resolution for *cis* and *thom* contact maps. One of the key factors defining the resolution of a Hi-C dataset is the sequencing depth. If we pick a resolution too high (i.e. bin size is too small) for a given sequencing depth, we end up with many empty pixels, and overall, read counts would be distributed over many more pixels and thus would be more prone to sampling noise. In addition, because the mean number of read counts decays with distance, more distant diagonals will have more empty pixels. Thus, to address the difficulty of determining the optimal resolution across an entire contact map, we try different bin sizes and determine the number of empty pixels along each diagonal. The optimal resolution would be the one where there are only a few empty pixels at diagonals-of-interest - i.e. diagonals containing TADs ($< \sim 100$ kb). In the case of untreated PnM cells, we looked at four resolutions: 1 kb, 2 kb, 4 kb, and 10 kb and plotted the fraction of non-zero pixels in diagonals as a function of the genomic separation or distance (Extended Data Fig. 14). To interpret these curves, we set a criterion wherein the best resolution was the finest resolution at which we still had more than 50% non-zero pixels at 100

kb separation (relative average TAD scale). By this criterion, a 4 kb resolution emerged as optimal for both *cis* and *trans* contact maps.”

6. Can the authors please clarify their claim on page 4, line 7 that “pairing can serve as a genome-wide regulatory mechanism”? What specifically is being regulated? Is this meant to be related to transvection and gene expression?

Response: We agree with the Reviewer and see how our wording was misleading, as we did not intend to imply any specific mechanism.

Excerpt (Introduction) P4, line 8: “... Thus, the question as to whether pairing can serve as a global mechanism for regulating and coordinating function, possibly facilitating transvection genome-wide, is drawing increasing attention.

Other author comments:

1. During our revision process, we received Reviewer comments for our companion paper (*Erceg et al*, NCOMMS-19-02486-T), which had been reviewed by an overlapping set of Reviewers. Those comments made clear that we had not sufficiently distinguished the objectives of this manuscript. Therefore, we have modified the introduction and abstract to emphasize the differences between the two manuscripts as well as highlight those findings that are unique to this manuscript:

Excerpt (Abstract) P2, line 2: “*Trans*-homolog interactions have been studied extensively in *Drosophila*, where homologs are paired in somatic cells, and transvection is prevalent. Nevertheless, the detailed structure of pairing and its functional impact have not been thoroughly investigated. Accordingly, we generated a diploid cell line from divergent parents and applied haplotype-resolved Hi-C, showing that homologs pair with varying precision genome-wide in addition to establishing *trans*-homolog domains and compartments. We also elucidated the structure of pairing with unprecedented detail, observing significant variation across the genome and revealing at least two forms of pairing: tight pairing, spanning contiguous small domains, and loose pairing, consisting of single larger domains. Strikingly, active genomic regions (A-type compartments, active chromatin, expressed genes) correlated with tight pairing, suggesting that pairing has a functional implication genome-wide. Finally, using RNAi and haplotype-resolved Hi-C, we show that disruption of pairing-promoting factors results in global changes in pairing, including the disruption of some interaction peaks.”

Excerpts (Introduction) P3, line 8: “...Although long considered relevant only to meiosis, *trans*-homolog interactions are now widely recognized for their capacity to affect gene function in *Drosophila*, where homologs are paired in somatic cells throughout nearly all of development (reviewed by (*McKee, 2004, Apte and Meller, 2012, Kassiss, 2012, Joyce et al., 2016, Fukaya and Levine, 2017*)). What remains unclear is the global impact of such interactions and our ability to comprehensively understand the structure of paired chromosomes. To tackle this issue,

we examine genome-wide maps of *trans*-homolog interactions in a newly established *Drosophila* hybrid cell line (PnM, XY diploid).”

P5, line 4: “...Here, we describe our work in examining the detailed architecture of pairing, using haplotype-resolved Hi-C to specifically target the pairing that occurs between homologous chromosomes. Haplotype-resolved Hi-C has been used to investigate *cis* interactions within mammalian genomes (*Selvaraj et al., 2013, Rao et al., 2014, Deng et al., 2015, Dixon et al., 2015, Minajigi et al., 2015, Darrow et al., 2016, Giorgetti et al., 2016, Du et al., 2017, Ke et al., 2017, Barutcu et al., 2018, Bonora et al., 2018, Tan et al., 2018*) and diploid homolog pairing in yeast (*Kim et al., 2017*) and, in our companion paper (*Erceg, AlHaj Abed, Goloborodko et al., 2018, bioRxiv*), we developed a general methodology for applying this approach that ensures minimal misassignment of reads and high stringency in the detection of pairing. Applied to mammalian and *Drosophila* embryos, this approach demonstrated pairing in the latter to be genome-wide and also provided a framework in which to consider pairing in terms of precision, proximity, and continuity. We further revealed a potential connection between pairing and the maternal-to-zygotic transition in early embryogenesis.”

P5, line 15: “In the current study, we shifted our focus to the fine structure of somatically paired homologs and, to that end took advantage of the greater homogeneity and higher pairing levels of *Drosophila* cell culture. In particular, we generated a diploid cell line from a hybrid cross and then applied haplotype resolved Hi-C, allowing us to achieve a high-resolution map of homolog pairing. This approach revealed *trans*-homolog domains, interaction peaks, and compartments as well as variation in the structure and precision of pairing, documenting an extensive interspersion of tightly paired regions with loosely paired regions across the genome. Excitingly, we also found a strong association between pairing and active chromatin, compartments, and gene expression. Our findings demonstrate a comprehensive and detailed view of the structure of homolog pairing and resolve the long-standing question of whether pairing can bear a genome-wide relationship to gene expression.”

2. While our paper was in review, a study by Rowley *et al*, 2019 investigating 3D genome organization and allelic interactions was published. In addition, a study by Mateo *et al*, 2019 demonstrated pairing of small regions at super-resolution at the BX-C locus. Thus, the revised manuscript either cites and/or discusses the findings of these publications, comparing them, when relevant, to our findings:

Excerpts (Introduction) P3 , line 8: “...Although long considered relevant only to meiosis, *trans*-homolog interactions are now widely recognized for their capacity to affect gene function in *Drosophila*, where homologs are paired in somatic cells throughout nearly all of development (reviewed by (*McKee, 2004, Apte and Meller, 2012, Kassis, 2012, Joyce et al., 2016, Fukaya and Levine, 2017*)). What remains unclear is the global impact of such interactions and our ability to comprehensively understand the structure of paired chromosomes. To tackle this issue, we examine genome-wide maps of *trans*-homolog interactions in a newly established *Drosophila* hybrid cell line (PnM, XY diploid). In particular, by taking advantage of the parent-specific single nucleotide variants (SNVs) in this cell line, we provide a global assessment of

different properties of homolog pairing, including how tightly apposed homologous chromosomes are and whether pairing is uniform across the genome. Furthermore, due to the sensitivity SNVs afforded our study, we assess how homolog proximity correlates with precision of alignment and with genome function.

P4, line 14: “...Recent studies have also used live imaging (*Lim et al., 2018*) as well as fluorescent *in situ* hybridization (FISH) achieving super-resolution to visualize pairing of genomic regions as small as a few kilobases to as large as megabases, wherein a single signal in a nucleus was interpreted as the paired state and two as the unpaired state (*Cattoni et al., 2017, Szabo et al., 2018, Cardozo Gizzi et al., 2019, Mateo et al., 2019*). Chromosome conformation capture technologies, such as Hi-C, have also been implemented in investigations of pairing in yeast (*Kim et al., 2017*). A recent study used read pairs representing interactions between identical Hi-C restriction fragments in a *Drosophila* cell line (Kc₁₆₇ cells, XXXX tetraploid) to tease out allelic interactions, such as between two homologs and between two sister chromatids (*Rowley et al., 2019*). This study reported an enhancement of allelic pairing in active genomic regions as well as an involvement of architectural proteins. In addition, consistent with the Cap-H2 component of condensin II being an anti-pairing factor (*Hartl et al., 2008*) and Slmb being a negative regulator of Cap-H2 (*Hartl et al., 2008, Joyce et al., 2012, Buster et al., 2013*), this study reported increased and decreased allelic interactions, respectively, in the absence of these factors.”

P5, line 4: “...Here, we describe our work in examining the detailed architecture of pairing, using haplotype-resolved Hi-C to specifically target the pairing that occurs between homologous chromosomes.”

Excerpt (Results) P8, line 1: “...We began by comparing the probability with which a locus will interact with another locus in *cis* as versus in *trans* at varying genomic separations (within a few kilobases and up to tens of megabases). We reasoned that, if pairing were maximally precise, tight, and continuous (‘railroad track’), our use of SNVs would reveal that any two loci would interact in *trans* nearly as often as they interact in *cis* regardless of genomic separation.”

Excerpt (Concluding remarks) P17, line 5: “...Using SNVs and our haplotype-resolved approach, we find that pairing is extensive, spanning a wide range of genomic distances, from as small as few kilobases to as large as tens of megabases, and includes *trans* domains, loops, and compartments. Furthermore, we observed two forms of pairing (Fig. 4b): a tighter, more precise form that can encompass many contiguous small domains paired at their boundaries and a looser, less precise form often corresponding to single domains flanked by tight pairing at the boundaries...”

REVIEWERS' COMMENTS:

Reviewer #1 (Remarks to the Author):

The authors fully addressed my previous comments. The analysis of architectural proteins at the pairing sites will provide an important insight into the mechanism underlying homolog pairing. I believe this study will be of an interest to the Nature Communication readership.

Reviewer #2 (Remarks to the Author):

The authors have done an excellent job addressing my previous concerns. I recommend publication of this work in Nature Communications.

Reviewer #3 (Remarks to the Author):

My comments have been sufficiently addressed.

I also thank the authors for clarifying how this manuscript differs from their companion paper, Erceg et al., and for updating the manuscript in regard to recent publications. This specific system, and the use of SNVs for resolving the haplotypes, is distinct from and has advantages over other approaches (e.g. Rowley et al., 2019) so I am very supportive of publication of this manuscript in Nature Communications.